# Learning to Optimize Quasi-Newton Methods

**Isaac C. Liao** *iliao@mit.edu*
*Research Lab for Electronics*
*MIT*

**Rumen R. Dangovski** *rumenrd@mit.edu*
*Research Lab for Electronics*
*MIT*

**Jakob N. Foerster** *jfoerster@cs.toronto.edu*
*Department of Engineering Science*
*University of Oxford*

**Marin Soljačić** *soljacic@mit.edu*
*Research Lab for Electronics*
*MIT*

**Reviewed on OpenReview:** *https://openreview.net/forum?id=Ns2X7Azudy*

## Abstract

Fast gradient-based optimization algorithms have become increasingly essential for the computationally efficient training of machine learning models. One technique is to multiply the gradient by a preconditioner matrix to produce a step, but it is unclear what the best preconditioner matrix is. This paper introduces a novel machine learning optimizer called LODO, which tries to online meta-learn the best preconditioner during optimization. Specifically, our optimizer merges Learning to Optimize (L2O) techniques with quasi-Newton methods to learn preconditioners parameterized as neural networks; they are more flexible than preconditioners in other quasi-Newton methods. Unlike other L2O methods, LODO does not require any meta-training on a training task distribution, and instead learns to optimize *on the fly* while optimizing on the test task, adapting to the local characteristics of the loss landscape while traversing it. Theoretically, we show that our optimizer approximates the inverse Hessian in noisy loss landscapes and is capable of representing a wide range of inverse Hessians. We experimentally verify that our algorithm can optimize in noisy settings, and show that simpler alternatives for representing the inverse Hessians worsen performance. Lastly, we use our optimizer to train a semi-realistic deep neural network with 95k parameters at speeds comparable to those of standard neural network optimizers.

## 1 Introduction

Many optimization algorithms like stochastic gradient descent (SGD) (Rosenblatt, 1958) and Adam (Kingma & Ba, 2014) have been widespread and successful in the rapid training of deep neural networks (Sun et al., 2019a). Fundamentally, this is a problem of minimizing a loss which is a function of a large vector containing the weights of the network. The time it takes to optimize a neural network is a bottleneck in machine learning. The more quickly a network can be trained, the more computational resources are saved, and therefore researchers have devoted great effort into creating new and faster optimizers. (Jain & Kar, 2017; Metz et al., 2020; Bernstein et al., 2020; Martens & Grosse, 2015a; Sun et al., 2019b)

We present a novel algorithm drawing from the field of *learning to optimize* (L2O) spearheaded by (Li & Malik, 2016) and (Andrychowicz et al., 2016). Namely, we use a meta-optimizer to online learn an optimization strategy which reaches lower losses with fewer steps. Our learned optimization strategy takes

the form of an online learned neural network representation of the local inverse Hessian of the loss which is meanwhile being used as a gradient preconditioner for quasi-Newton optimization. Unlike in the Newton method, matrix-vector multiplication by the inverse Hessian estimate is cheap for our neural representation, allowing our optimizer to train much larger neural networks than the Newton method can. Unlike other L2O algorithms which learn to optimize *before* optimization (Chen et al., 2021), our algorithm "learns to optimize *during* optimization" (LODO for short) without any L2O meta training time on a curated training task distribution, and is instead applied directly and immediately to the desired optimization problem with no preparation. This way, LODO learns during training how to exploit any local curvatures of the loss landscape within each use case. Our work on LODO primarily aims to foster further study on the blending of quasi-Newton and L2O methodologies, specifically by combining the step efficiency of the Newton method with the gradient-based learning capabilities of neural representations of matrices. We present theoretical work to show that this blending gives rise to desirable optimization behavior in nonlinear and stochastic problems, and experiments to support this theory. We target the problem of training model architectures with lots of parameter sharing, such as convolutional networks, where the time taken for inference within the training loop drastically outweighs the time taken by the optimizer.

**Claims and Evidence.** We show theoretically and experimentally that a simplified version of LODO correctly learns the inverse Hessian in a stochastic convex setting. Next, we show theoretically that LODO's inverse Hessian representation is highly expressive, and experimentally that simpler, less expressive alternatives perform worse. Finally, we demonstrate the use of LODO in an autoregressive image generation task and in image classification. This paper serves as a stepping stone in the development of meta-training-free online L2O.

The remainder of this paper is structured as follows. Section 2 discusses relevant background and contributions in optimization and L2O. Section 3 shows how LODO works. Section 4 provides theoretical justification for our design of LODO. Section 5 shows experiments which explore what makes LODO work and why. Section 6 discusses our findings and Section 7 summarizes them.

## 2 Related Work

Research into the construction of faster optimizers has mostly fallen under two branches of work. The older branch attempts to endow SGD with adaptive capabilities, often through modifications involving calculations of means and variances of the gradient using exponential moving averages (EMAs). These values are then combined to create a preconditioner matrix which is then multiplied by the gradient to choose a step. RMSprop (Hinton et al., 2012) and Adam use the mean to induce momentum with a variance-based diagonal preconditioner to normalize the step size. LARS (You et al., 2017) modifies the preconditioner calculation to normalize layer-wise, while Yogi (Zaheer et al., 2018) modifies the variance accumulation to control increases in effective learning rate in a slower manner.

Some methods such as the Newton method and natural gradient descent (Martens & Grosse, 2015c; George et al., 2018) precondition the gradient with adaptive estimates of the inverse Hessian and the inverse Fisher information matrices, respectively. These methods converge quickly but can be vulnerable to gradient noise and/or impractical to implement due to the resources spent in calculating and/or inverting the high dimensional matrices involved. For example, a prominent natural gradient based optimizer is K-FAC (Martens & Grosse, 2015b), which preconditions using a Kronecker-factorized inverse Fisher estimator, and requires numerous expensive matrix inversions. For the Newton method, many researchers have developed approximations of the algorithm—called quasi-Newton methods—which have reduced time and memory complexity, such as L-BFGS (Nocedal & Wright, 1999) and variants better suited to the stochasticity and structure present in machine learning optimization problems (Schraudolph et al., 2007; Parker-Holder et al., 2020; Goldfarb et al., 2020; Park & Oliva, 2019; Yao et al., 2021).

In a quasi-Newton method, an approximate solution $\boldsymbol{x}_t \in \mathbb{R}^n$ is refined by $\boldsymbol{x}_{t+1} = \boldsymbol{x}_t - \tilde{\alpha} \boldsymbol{G}_t \boldsymbol{g}_t$ for some learning rate $\tilde{\alpha} > 0$, where $\boldsymbol{G}_t \approx (\nabla^2_{\boldsymbol{x}_t} f(\boldsymbol{x}_t))^{-1} \in \mathbb{R}^{n \times n}$ is some approximation of the inverse Hessian and $\boldsymbol{g}_t = \nabla_{\boldsymbol{x}_t} f(\boldsymbol{x}_t) \in \mathbb{R}^n$ is the gradient computed by backpropagation from the loss $f : \mathbb{R}^n \to \mathbb{R}$. $\boldsymbol{G}_t$ is usually

computed from a past history of $(\boldsymbol{x}_{t-i}, \boldsymbol{g}_{t-i})$ pairs. $\tilde{\alpha} = 1$ produces the exact solution for quadratic $f$, so we assume that $\tilde{\alpha} = 1$ from this point on.

The big difference which distinguishes LODO from other quasi-Newton methods is that its preconditioner is learned by a meta-optimizer through a method known as "hypergradient descent" (Baydin et al., 2017), rather than estimated using a hardcoded formula like the methods listed above. While past hypergradient methods use low-rank (Moskovitz et al., 2019), diagonal (Amid et al., 2022; Baydin et al., 2017), or Kronecker-factorized (Bae et al., 2022; Goldfarb et al., 2020) preconditioners, LODO uses an entirely different, neural network-based style of preconditioner, which is more expressive than low-rank and diagonal preconditioners while requiring lower time complexity than K-FAC. LODO is distinguished from the above methods in two aspects: the novel preconditioner, and the meta-learning approach for training the preconditioner. Our theory is focused on these two aspects of LODO. We present evidence of the theory in controlled experiments, of which (Amid et al., 2022; Baydin et al., 2017) are our ablations. We underscore that our objective is not to compare against all hypergradient methods but to focus on the *theory* of LODO and its validation in controlled experiments.

More recently, a subfield of meta-learning known as learning to optimize (L2O) has shown that deep networks can themselves be trained offline to perform optimization, at a speed which exceeds that of popular traditional optimizers, in hopes of further accelerating training procedures for other deep neural networks. Andrychowicz et al. (2016) (whose optimizer we will call "L2LBGDBGD") and Li & Malik (2016; 2017) were among the first to successfully train neural networks to transform gradients into high quality steps, by backpropagating through unrolled and truncated training loops to tune the optimizer. Since then, many other variations of this idea have successfully produced optimizers exceeding the speed of common optimizers for narrow ranges of machine learning models (Metz et al., 2018), though theoretical analysis of these learned optimizers tends to be difficult and scarce. A major goal of L2O research is to learn a single optimizer which can generalize to be able to train a wide variety of machine learning models with speed. (Lv et al., 2017)

Two issues also prevent L2O optimizers from being rapidly developed experimentally. Firstly, a carefully chosen "task distribution" for optimization practice is required for the meta-training of the L2O optimizer, playing the role analogous to the "dataset" but for learning how to optimize rather than to classify or to predict. These tasks are difficult to curate because the issue of generalization error applies; we need the test task to be similar to the training task distribution. Secondly, meta-training is prohibitively costly in that it involves nested training loops, which run much slower than a single training loop like in traditional learning (Metz et al., 2019). While other L2O methods are burdened by choice of task distribution and lengthy meta-training, LODO avoids these issues by learning to optimize online directly on the test task.

## 3 How LODO Works

Our algorithm approximates the inverse Hessian in a quasi-Newton method using a matrix $\boldsymbol{G}_t = \boldsymbol{G}(\boldsymbol{\theta}_t) \in \mathbb{R}^{n \times n}$ parameterized by a vector $\boldsymbol{\theta}_t$ of weights learned over time $t$, described later in this section. After every step $t \leftarrow t + 1$ using the formula

$$\boldsymbol{x}_{t+1} = \boldsymbol{x}_t - \boldsymbol{G}(\boldsymbol{\theta}_t)\boldsymbol{g}_t, \tag{1}$$

the loss $\ell_{t+1} = f(\boldsymbol{x}_{t+1})$ is computed. Then the new gradient $\nabla_{\boldsymbol{x}_{t+1}}\ell_{t+1}$ in $\boldsymbol{x}_{t+1}$ is computed through backpropagation as usual, but for LODO we continue backpropagation through Equation 1 until we find the "hypergradient" $\nabla_{\boldsymbol{\theta}_t}\ell_{t+1}$ in the optimizer weights $\boldsymbol{\theta}_t$, which allows us to update the optimization strategy as well. In this manner, $\boldsymbol{\theta}_t$ is trained such that the quasi-Newton method tries to minimize the loss upon taking a single step.

In the L2O interpretation of such an algorithm, this would be equivalent to unrolling the inner loop optimization for only one step, causing severe truncation bias; the optimizer learns to greedily optimize within too short of a time horizon, thus suffering in the long term (Metz et al., 2019). As a countermeasure, we take inspiration from the Momentum modification to SGD, and replace $\boldsymbol{g}_t$ by a gradient EMA $(1-\beta)\sum_{\tau=0}^{\infty}\beta^{t-\tau}\boldsymbol{g}_\tau$ for use in Equation 1. This changes Equation 1 to $\boldsymbol{x}_{t+1} = \boldsymbol{x}_t - (1-\beta)\boldsymbol{G}(\boldsymbol{\theta}_t)\sum_{\tau=0}^{\infty}\beta^{t-\tau}\boldsymbol{g}_\tau$ instead, for some decay parameter $\beta$, producing our LODO algorithm, summarized in Algorithm 1. While we do not theoretically analyze the effect of momentum accumulation on LODO, Section 5.3.2 shows experimentally that momentum is beneficial in practice. We leave any theoretical analysis of momentum for future work.

---

**Algorithm 1** Learning to Optimize During Optimization (LODO)

---

**Require:** $f : \mathbb{R}^n \to \mathbb{R}$: Function to minimize.
**Require:** $\boldsymbol{x}_0 \in \mathbb{R}^n$: Initialization.
**Require:** $\alpha \in \mathbb{R}$: Meta-learning rate (default 0.001),
**Require:** $\alpha_0 \in \mathbb{R}$: Initial learning rate (default 1.0),
**Require:** $0 \leq \beta < 1$: Momentum (default 0.9),
   $t \leftarrow 0$                                                  ▷ Start time
   $\boldsymbol{\theta}_0 \leftarrow$ random initialization                 ▷ Initialization for $\boldsymbol{G}$ neural network
   $\boldsymbol{m}_0 \leftarrow \boldsymbol{0}$                                  ▷ Initialize momentum
   **while** not converged **do**
      $\boldsymbol{x}_{t+1} \leftarrow \boldsymbol{x}_t - \boldsymbol{G}(\boldsymbol{\theta}_t)\boldsymbol{m}_t$         ▷ Pick a step using $\boldsymbol{G}$ with Eqs. (1) and (2)
      $\ell_{t+1} \leftarrow f(\boldsymbol{x}_{t+1})$                      ▷ Compute loss after step
      $\boldsymbol{\theta}_{t+1} \leftarrow \boldsymbol{\theta}_t + \text{Adam}(\nabla_{\boldsymbol{\theta}_t}\ell_{t+1})$     ▷ Tune the $\boldsymbol{G}$ model to pick better steps
      $\boldsymbol{m}_{t+1} \leftarrow \beta\boldsymbol{m}_t + (1-\beta)\nabla_{\boldsymbol{x}_{t+1}}\ell_{t+1}$              ▷ Update momentum
      $t \leftarrow t + 1$                                    ▷ Increment time
   **end while**
   **return** $\boldsymbol{\theta}_t$

---

Our parameterization of $\boldsymbol{G}(\boldsymbol{\theta})$ is uniquely inspired by the FFT-style Efficient Unitary Neural Network (EUNN) (Jing et al., 2017) designed for parameter-efficient representations of unitary matrices. We replace the fixed distance interactions by random interactions and force the result to be symmetric[1], by choosing $\boldsymbol{G}(\boldsymbol{\theta})$ to be the following product of many matrices:

$$\boldsymbol{G}(\boldsymbol{\theta}) = \alpha_0 \begin{pmatrix} \boldsymbol{I} & \boldsymbol{0} \end{pmatrix} \tilde{\boldsymbol{G}}(\boldsymbol{\theta})^T \tilde{\boldsymbol{G}}(\boldsymbol{\theta}) \begin{pmatrix} \boldsymbol{I} \\ \boldsymbol{0} \end{pmatrix}$$

$$\tilde{\boldsymbol{G}}(\boldsymbol{\theta}) = \prod_{i=1}^{N} \boldsymbol{B}(\boldsymbol{\theta}^{(i)})\boldsymbol{P}_i \tag{2}$$

where $\alpha_0$ is a fixed initial learning rate, $\boldsymbol{P}_i$ are randomly selected permutation matrices, and $\boldsymbol{B}(\boldsymbol{\theta}^{(i)})$ are block-diagonal matrices whose block contents are listed in a subsection $\boldsymbol{\theta}^{(i)}$ of the parameter vector $\boldsymbol{\theta} = (\boldsymbol{\theta}^{(1)}, \dots \boldsymbol{\theta}^{(N)})$, as illustrated in Figure 1. Every block is size $k \times k$ for some chosen $k$ (we use 4 for our setup). The $\begin{pmatrix} \boldsymbol{I} & \boldsymbol{0} \end{pmatrix}$ and $\begin{pmatrix} \boldsymbol{I} & \boldsymbol{0} \end{pmatrix}^T$ matrices are $n \times \tilde{n}$ and $\tilde{n} \times n$ respectively, where $\tilde{n}$ is some integer chosen to be larger than $n$ (we use $\tilde{n} = \lfloor 2n/k \rfloor k$ for our setup), and all other matrices are $\tilde{n} \times \tilde{n}$. By initializing each block matrix in $\boldsymbol{B}(\boldsymbol{\theta}^{(i)})$ to a random orthogonal matrix, we ensure that $\boldsymbol{G}(\boldsymbol{\theta}) = \alpha_0 \boldsymbol{I}$ upon initialization, despite the input information diffusing and mixing with itself as more of the first $N$ block matrices are applied (for our setup we choose $N = 16$), normalizing hypergradient magnitudes to facilitate the initial meta-learning of $\boldsymbol{\theta}$. In effect, this causes LODO to begin with the strategy of SGD with the same learning rate in all directions.

The product $\boldsymbol{G}(\boldsymbol{\theta})$ intentionally resembles the operation performed by a deep neural network with $N$ layers, millions of hidden neurons arranged in inverted bottleneck style, very sparse connections, and no activation functions. We intend for the random connections between neurons to bring expander graph properties to the computation graph of the neural network, such that input signals can diffuse and self-mix quickly without travelling long distances within the computation graph.

Since we expect the receptive field of output nodes to grow exponentially with depth, it takes $O(\log n)$ layers for all of the $n$ inputs to interact with each other, so we intend for the depth $N$ to be not much larger than $O(\log n)$. Applying permutations and block diagonal matrices both require $O(n)$ time and memory, so matrix-vector multiplication with the inverse Hessian estimate $\boldsymbol{G}(\boldsymbol{\theta})$ costs $O(n \log n)$ time and memory. We intend for LODO to be used to train machine learning architectures of high parameter reuse such as CNNs and transformers, such that this $O(n \log n)$ cost of LODO is small in comparison to the forward and backward passes of the trained model itself.

---

[1]This is because the true inverse Hessian is also symmetric.

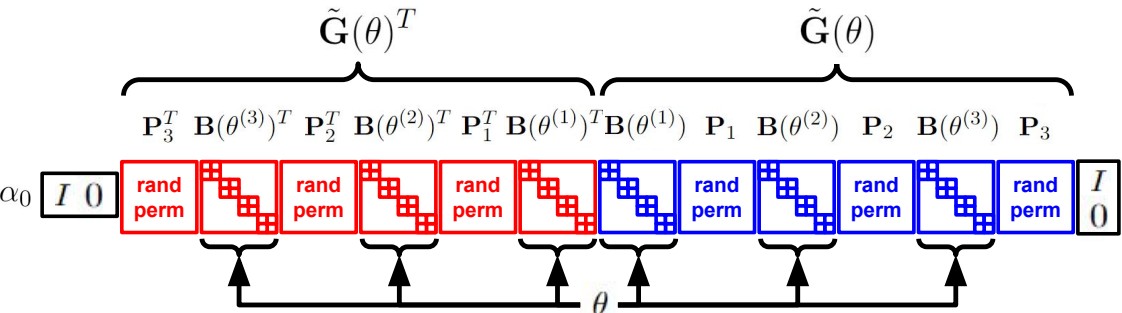

Figure 1: Visualization of LODO's matrix structure of $\boldsymbol{G}(\boldsymbol{\theta})$ from Equation (2) for the approximation of the inverse Hessian. Reduced to a depth of 3, block size 2, and size $\tilde{n} = 8$ matrices for illustration.

## 4 Theoretical Properties of LODO

In this section, we mainly present two theoretical results: in Section 4.1 about desirable preconditioner learning dynamics, and in Section 4.3 about preconditioner expressiveness; we also show that LODO repels saddle points in Section 4.2 and there is further analysis from the literature for the preconditioner approximation error function in Appendix D.

### 4.1 Inverse Hessian Learning Dynamics

We first motivate that under certain conditions, LODO learns the Hessian over time. This result allows LODO to come arbitrarily close to the true inverse Hessian despite noise, which makes LODO more noise-tolerant than most other quasi-Newton methods, whose preconditioners may vary wildly in the presence of noise. We assume that the loss function to minimize is quadratic with fixed positive-definite Hessian $\boldsymbol{H}$, where the minimum $\boldsymbol{x}_t^*$ is perturbed by noise $\boldsymbol{s}_t$ at every time step $t$,

$$\ell_t = f_t(\boldsymbol{x}_t) = \frac{1}{2}(\boldsymbol{x}_t - \boldsymbol{x}_t^*)^T \boldsymbol{H}(\boldsymbol{x}_t - \boldsymbol{x}_t^*) \tag{3}$$

$$\boldsymbol{x}_{t+1}^* = \boldsymbol{x}_t^* + \boldsymbol{s}_t. \tag{4}$$

The perturbations $\boldsymbol{s}_t$ are i.i.d. from some light tailed distribution of zero mean, for example a multivariate normal distribution $\mathcal{N}(\boldsymbol{s}_t; 0, \boldsymbol{\Sigma})$.

For this section, we restrict our analysis to a simplified version of LODO, though the experiment of Section 5.1 supports similar findings for the full version of LODO as well. In our simplified version, the inverse Hessian is approximated by a full dense matrix $\boldsymbol{G}(\boldsymbol{\theta}_t)$ rather than the compositionally sparse architecture of Equation 2, so this result is relevant to many hypergradient methods and is not restricted to LODO. We will also update $\boldsymbol{\theta}_t$ using SGD of learning rate $\alpha$ instead of Adam; and we will feed the gradients directly into the $\boldsymbol{G}$ network using Equation 1 rather than accumulating them into EMAs first, ie. $\beta = 0$.

Under these conditions, the gradient of loss with respect to both $\boldsymbol{x}_t$ and $\boldsymbol{\theta}_t$ can be explicitly solved for. LODO turns out to follow the training dynamics

$$\boldsymbol{A}_{t+1} = \boldsymbol{A}_t - \alpha \boldsymbol{H} \boldsymbol{b}_{t+1} \boldsymbol{b}_t^T \boldsymbol{H}^2 \tag{5}$$

$$\boldsymbol{b}_{t+1} = \boldsymbol{A}_t \boldsymbol{b}_t - \boldsymbol{s}_t \tag{6}$$

with the change of variables to $\boldsymbol{A}_t = \boldsymbol{I} - \boldsymbol{G}(\boldsymbol{\theta}_t)\boldsymbol{H}$ the inverse Hessian approximation error and $\boldsymbol{b}_t = \boldsymbol{x}_t - \boldsymbol{x}_t^*$ the quadratic minimum approximation error, to replace $\boldsymbol{G}(\boldsymbol{\theta}_t)$ and $\boldsymbol{x}_t$ in the analysis. A full derivation of Equations 5 and 6 is in Appendix A.1. Assuming that the learning rate $\alpha$ is low enough and the error $\boldsymbol{A}_t$ begins small enough to have spectral norm less than 1, $\boldsymbol{A}_t$ must evolve very slowly while $\boldsymbol{b}_t$ comparatively quickly exponentially decays to a fixed distribution $\mathcal{B}_t$ dependent on $\boldsymbol{A}_t$ as determined by Equation 6. Furthermore, since $\boldsymbol{A}_t$ moves slowly, $\mathcal{B}_t \approx \mathcal{B}_{t_0}$ so long as $t - t_0$ for some reference time $t_0$ is not too large.

Thus, to determine the direction of slow movement of $\boldsymbol{A}_t$ (we seek to show that it moves towards zero), we can use Equation 5 as though $\boldsymbol{b}_t$ were sampled i.i.d. from $\mathcal{B}_0$. Appendix A.2 gives more rigorous justification for this approximation. Mathematically, we keep Equation 5 but approximate Equation 6 with

$$\boldsymbol{b}_{t+1} = \boldsymbol{A}_{t_0}\boldsymbol{b}_t - \boldsymbol{s}_t. \tag{7}$$

By recursively substituting Equation 7 into itself and then into Equation 5, the total eventual contribution of all $\boldsymbol{s}_\tau$ for $t_0 \leq \tau \leq t$ to the slow movement of $\boldsymbol{A}_t$ is then

$$\alpha \boldsymbol{H} \boldsymbol{A}_{t_0} \left( \sum_{n=0}^{\infty} {\boldsymbol{A}_{t_0}}^n \left( \sum_{\tau=t_0}^{t} \boldsymbol{s}_\tau \boldsymbol{s}_\tau^T \right) ({\boldsymbol{A}_{t_0}}^n)^T \right) \boldsymbol{H}^2 \tag{8}$$

with some extra doubly summed terms for with pairwise products between $\boldsymbol{s}_{\tau_1}$ and $\boldsymbol{s}_{\tau_2}$ ($\tau_1 \neq \tau_2$), and some singly summed terms for products between $\boldsymbol{b}_{t_0}$ and $\boldsymbol{s}_\tau$. In the long term (but with low enough $\alpha$ to retain the approximation of Equation 7), the average contribution of each $\boldsymbol{s}_\tau$ towards the total of Expression 8 then converges to

$$\alpha \boldsymbol{H} \boldsymbol{A}_{t_0} \left( \sum_{n=0}^{\infty} {\boldsymbol{A}_{t_0}}^n \mathbb{E}\left[ \boldsymbol{s}_\tau \boldsymbol{s}_\tau^T \right] ({\boldsymbol{A}_{t_0}}^n)^T \right) \boldsymbol{H}^2, \tag{9}$$

where pairwise products of $\boldsymbol{s}_i$ and $\boldsymbol{s}_j$ for $i \neq j$ and other terms with $\boldsymbol{b}_{t_0}$ are negligible due to mutual independence and small $\alpha$. Expression 9 is then the step direction for the Euler method for approximating the solution to the differential equation

$$\frac{\mathrm{d}\boldsymbol{A}}{\mathrm{d}t} = -\boldsymbol{H}\boldsymbol{A} \sum_{n=0}^{\infty} \boldsymbol{A}^n \mathbb{E}\left[ \boldsymbol{s}\boldsymbol{s}^T \right] (\boldsymbol{A}^n)^T \boldsymbol{H}^2, \tag{10}$$

which can be shown to cause $\boldsymbol{A}$ to flow towards zero as desired.[2] One might ask how the approximation made by Equation 7 by taking $\boldsymbol{A}_t \approx \boldsymbol{A}_{t_0}$ continues to hold while $\boldsymbol{A}$ flows to zero. When given a time $t_0$, this approximation is only necessary to reach the conclusion that over short distances, $\boldsymbol{A}$ has the long-term flow rate/direction of Equation 10. After every Euler step on Equation 10, we may reinstantiate $t_0$ to the new value of $t$, such that the accuracy of the approximation $\boldsymbol{A}_t \approx \boldsymbol{A}_{t_0}$ is fully restored for Equation 7 to be accurate, and Equation 10 can be reached once again so the next Euler step can be taken. It is in this way that the movement of $\boldsymbol{A}$ is described by Equation 10.

The flow of $\boldsymbol{A}$ towards zero implies that $\boldsymbol{G}(\boldsymbol{\theta}_t)$ approaches $\boldsymbol{H}^{-1}$, meaning that the inverse Hessian approximation improves over time. Therefore, it is reasonable to believe that LODO learns better inverse Hessians over time, given small enough learning rate and good initialization. The Frobenius norm of the error decays faster when magnitudes/norms of $\boldsymbol{H}$ and $\mathbb{E}\left[ \boldsymbol{s}\boldsymbol{s}^T \right]$ are higher, indicating that both curvature of the Hessian and noisy movement of the quadratic bowl's minimum are good for learning the Hessian for fixed $\alpha$, as long as the approximations required for the above analysis stay accurate. An interpretation of this phenomenon is that the noise $\boldsymbol{s}$ in every direction provides a training signal for LODO to learn that direction's curvature and is amplified by the Hessian $\boldsymbol{H}$.

---

[2]$\boldsymbol{A}$ flows towards zero because by substituting the eigendecomposition $\boldsymbol{H} = \boldsymbol{U}\boldsymbol{D}\boldsymbol{U}^T$ where $\boldsymbol{U}\boldsymbol{U}^T = \boldsymbol{I}$, and using $\boldsymbol{B} = \boldsymbol{U}^T\boldsymbol{A}\boldsymbol{U}$, we can show that the norm of $\boldsymbol{B}\boldsymbol{D}^{-1}$ decreases over time,

$$\frac{\mathrm{d}\|\boldsymbol{B}\boldsymbol{D}^{-1}\|_F^2}{\mathrm{d}t} = \frac{\mathrm{d}}{\mathrm{d}t}\mathrm{tr}\left( \boldsymbol{D}^{-2}\boldsymbol{B}^T\boldsymbol{B} \right) \tag{11}$$

$$= -2\mathrm{tr}\left( \boldsymbol{B}^T\boldsymbol{D}\boldsymbol{B} \sum_{n=0}^{\infty} \boldsymbol{B}^n \boldsymbol{U}^T \mathbb{E}\left[ \boldsymbol{s}\boldsymbol{s}^T \right] \boldsymbol{U}(\boldsymbol{B}^n)^T \right) \tag{12}$$

$$= -2\left\| \boldsymbol{D}^{\frac{1}{2}}\boldsymbol{B} \left( \sum_{n=0}^{\infty} \boldsymbol{B}^n \boldsymbol{U}^T \mathbb{E}\left[ \boldsymbol{s}\boldsymbol{s}^T \right] \boldsymbol{U}(\boldsymbol{B}^n)^T \right)^{\frac{1}{2}} \right\|_F^2 \tag{13}$$

$$\leq 0 \tag{14}$$

where the strict equality is only satisfied for $\boldsymbol{A} = 0$.

## 4.2  Saddle Point Repulsion

We may also produce another analysis for quadratic loss functions of Hessian $\boldsymbol{H}$ with a direction of negative curvature, to show that LODO repels saddle points. Again, we restrict our analysis by omitting momentum as in Section 4.1, but here we keep the original architecture of $\boldsymbol{G}(\boldsymbol{\theta}_t)$ from Equation 2 and the Adam meta-optimizer, and merely set the meta-learning rate $\alpha$ to be negligibly small. From Equation 2, $\boldsymbol{G}(\boldsymbol{\theta}_t)$ is a positive-definite symmetric matrix in most cases of $\boldsymbol{\theta}_t$, and is effectively fixed due to small $\alpha$. Then, $\boldsymbol{G}(\boldsymbol{\theta}_t)\boldsymbol{H}$ must have some negative eigenvalue $\lambda$ with eigenvector $\boldsymbol{y}$ (see Appendix B for proof), such that the matrix $\boldsymbol{A} = \boldsymbol{I} - \boldsymbol{G}(\boldsymbol{\theta}_t)\boldsymbol{H}$ has an eigenvalue of norm greater than 1, meaning that the displacement $\boldsymbol{b} = \boldsymbol{x} - \boldsymbol{x}^*$ from the center of the saddle $\boldsymbol{x}^*$ blows up exponentially in the $\boldsymbol{y}$ direction. Since $\boldsymbol{G}(\boldsymbol{\theta}_t)\boldsymbol{H}\boldsymbol{y} = \lambda\boldsymbol{y}$, left multiplying by $\boldsymbol{y}^T\boldsymbol{G}(\boldsymbol{\theta}_t)^{-1}$ further shows that $\boldsymbol{y}^T\boldsymbol{H}\boldsymbol{y} < 0$, implying that the direction $\boldsymbol{y}$ of saddle point repulsion is one of negative curvature, which decreases the loss.

## 4.3  Preconditioner Expressiveness

In this section, we seek to show that our parameterization of $\boldsymbol{G}(\boldsymbol{\theta})$ is highly expressive; LODO may represent a wider range of inverse Hessians than many other methods. In fact, we show that with an increase in parameter count by only a logarithmic factor in the task dimension, our fixed parameterization of $\boldsymbol{G}(\boldsymbol{\theta})$ can exactly replicate any possible parameterizations $\tilde{\boldsymbol{F}}^T\tilde{\boldsymbol{F}}$ where vector multiplication with a matrix $\tilde{\boldsymbol{F}}$ is computable with almost any arbitrary sparse linear neural network of a given size. Our parameterization is thus capable of replicating scalar and diagonal preconditioners from (Baydin et al., 2017) and (Amid et al., 2022) with only $O(\tilde{n}\log\tilde{n})$ parameters, and low rank preconditioners (Moskovitz et al., 2019) with only $O(\tilde{n}\log^2\tilde{n})$ parameters. Definition 4.1 creates a function $\epsilon$ to characterize the mixing rate of random transpositions when trying to shuffle lists, while Theorem 4.2 uses this $\epsilon$ function to lower bound the probability that the $\tilde{\boldsymbol{G}}(\boldsymbol{\theta})$ network in LODO can represent other linear neural networks when randomly sampling over the permutations $\boldsymbol{P}_i$ in Equation 2.

**Definition 4.1.** Uniformly sample a sequence of $N\tilde{n}/2$ transpositions of two out of $\tilde{n}$ elements, for integers $\tilde{n}/2 \in \mathbb{N}$ and $N \in \mathbb{N}$, with the condition that every successive block of $\tilde{n}/2$ transpositions commutes internally (transpositions can be rearranged within a block). We replace each transposition with the identity operation with probability $1/2$, and then compose the sequence of transpositions/identities to form a permutation. Then, we define the function $\epsilon(N\tilde{n}/2, \tilde{n})$ such that the expected entropy of this permutation, given the original sequence but not given the locations where identities were placed, is $\log\tilde{n}! - \epsilon(N\tilde{n}/2, \tilde{n})$.

**Theorem 4.2.** *Uniformly sample permutations $\boldsymbol{P}_i$ and create block-diagonal matrices $\boldsymbol{B}(\boldsymbol{\theta}^{(i)})$ where every block is $2 \times 2$, and whose block contents are listed by the parameters $\boldsymbol{\theta}^{(i)}$. Use these to construct the LODO subnetwork $\tilde{\boldsymbol{G}}(\boldsymbol{\theta})$ as in Equation 2 with some depth $N$ and hidden dimension $\tilde{n}$. Construct any linear neural network $\tilde{\boldsymbol{F}}$ with input dimension, output dimension, number of connections per layer at most $\tilde{n}$, at most $k$ incoming and at most $k$ outgoing connections for every neuron, depth $d$, and otherwise any arrangement of connections. Then, there is a probability of at least*

$$1 - \tilde{n}!N\sqrt{\frac{1}{2}\epsilon\left(\frac{\tilde{n}N}{4d(\lceil\log_2 k\rceil + 1)}, \tilde{n}\right)} \tag{15}$$

*that $\tilde{\boldsymbol{G}}(\boldsymbol{\theta}) = \tilde{\boldsymbol{F}}$ for some $\boldsymbol{\theta}$.*

We provide a constructive proof in Appendix C.

We believe that random transpositions in the style of Definition 4.1 are a quick way to shuffle lists via transposition, since the Cayley graph over the symmetric group generated by all transpositions has good expansion properties (Konstantinova & Kravchuk, 2022). In other words, we hypothesize that for large $N$ and $\tilde{n}$, we have $\epsilon(N\tilde{n}/2, \tilde{n}) \approx c_1\tilde{n}e^{-c_2 N\tilde{n}/2}$ for some positive constants $c_1$ and $c_2$. This would imply that the probability that $\tilde{\boldsymbol{G}}(\boldsymbol{\theta})$ can represent all possible $\tilde{\boldsymbol{F}}$ is at least approximately

$$1 - \tilde{n}!N\sqrt{\frac{c_1\tilde{n}}{2}\exp\left(\frac{-c_2\tilde{n}N}{4d(\lceil\log_2 k\rceil + 1)}\right)} \tag{16}$$

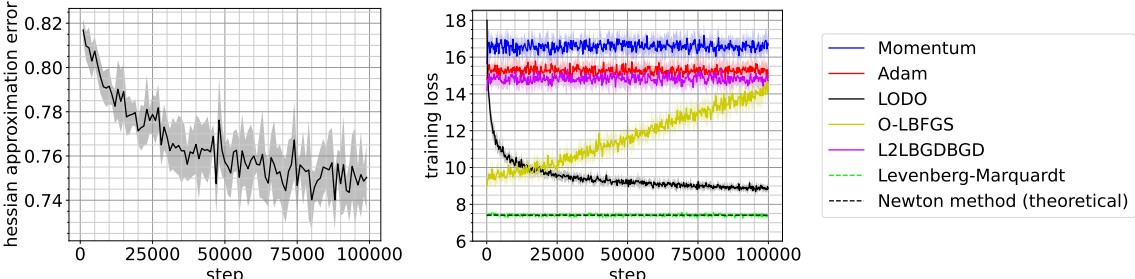

Figure 2: **Left:** LODO's average inverse Hessian approximation error $\sigma = \sqrt{||\boldsymbol{I} - \boldsymbol{G}(\boldsymbol{\theta}_t)\boldsymbol{H}||_F^2/n}$ on the noisy quadratic bowl task of Section 5.1. $\sigma^2$ is measured by the unbiased estimator $\frac{1}{100}\sum_{i=1}^{100}||(\boldsymbol{I} - \boldsymbol{G}(\boldsymbol{\theta}_t)\boldsymbol{H})\boldsymbol{v}_i||_2^2$ with random independent unit vectors $\boldsymbol{v}_i$. $\sigma = 1$ when $\boldsymbol{G}(\boldsymbol{\theta}_t)$ is trivially zero and $\sigma = 0$ when $\boldsymbol{G}(\boldsymbol{\theta}_t) = \boldsymbol{H}^{-1}$ as desired. Ordinarily, Theorem 4.2 would dictate that $\boldsymbol{G}(\boldsymbol{\theta}_t) = \boldsymbol{H}^{-1}$ is reachable for some $\theta_t$, but $\theta_t$ is nowhere near overparameterized enough for the theorem to apply, so we do not reach even close to $\sigma = 0$. Despite this, $\sigma$ still decreases over time. **Right:** Average training loss learning curves. The dotted line shows the theoretically best possible loss using Newton's method. Better optimizers maintain lower losses after infinite steps, since for this task, loss is introduced over time and the optimizer serves to quickly suppress it. Solid lines indicate methods whose time complexities allow for scaling to large neural networks (ie. $O(n \log n)$ or less,) while dotted lines indicate methods which are too expensive for this.

which can be made to be above $1 - (n!)^c$ for any constant $c$ by using sufficient depth $N$ which goes like $N \propto d(\log_2 k)\log \tilde{n}$, due to Stirling's approximation. Thus we believe that by only being $O((\log_2 k)\log \tilde{n})$ times deeper than $\tilde{\boldsymbol{F}}$, we can make it very likely that our model $\tilde{\boldsymbol{G}}(\boldsymbol{\theta})$ can represent all possible $\tilde{\boldsymbol{F}}$.

# 5 Experiments with LODO

We present here a number of tasks which provide experimental evidence for the theoretical results we claim, though we test a variety of optimizers on these tasks. Appendix E explains how we tuned each optimizer's hyperparameters for each task. Optimizers were used 8 times for every experiment to ensure reproducibility of results by randomly sampling over the permutations $\boldsymbol{P}_i$ in Equation 2 with different randomization seeds, unless otherwise stated. Error margins in Figures and intervals in tables indicate $\pm 1$ standard deviation across the the 8 runs. Our timing setup is described in Appendix H.

## 5.1 Noisy Quadratic Bowl

In Figure 2 and Table 1 we use various optimizers to track the minimum of a quadratic bowl of fixed true Hessian $\boldsymbol{H}$ as its minimum is perturbed by noise at every step, to demonstrate that LODO correctly learns its inverse Hessian representation as claimed in Section 4.1. The setup of the noisy quadratic bowl is the same as in Section 4.1 and details are provided in Appendix F.1. We interpret an optimizer to be better if it can maintain a lower loss in the infinite step limit, since the error builds at a constant rate over time and the optimizer's role is to react and correct it as quickly as possible. We tested each optimizer by using it to track the minimum of the moving quadratic bowl over 100k steps. Learning curves in Figure 2 and losses in Table 1 show that LODO tracks the minimum more accurately than other optimizers, and that an estimator of the inverse Hessian approximation error $||\boldsymbol{I} - \boldsymbol{G}(\boldsymbol{\theta}_t)\boldsymbol{H}||_F^2/n$ decays over time. Other optimizers underperform because their preconditioners are less expressive, being either diagonal or low-rank and thus unable to capture the off-diagonal elements or non-top-few singular values of the true inverse Hessian, leading to a higher loss.

**Sensitivity to Noise.** In Table 7 and Figure 8 of the Appendix, we test the performance of LODO in the presence of noise of isotropic variance $v$ no longer fixed to 1. While some other optimizers' noise-rescaled training losses $\ell/v$ (e.g. RMSProp, Yogi and Adam) are heavily dependent on the noise covariance $v$, LODO's noise-rescaled training loss does not, and LODO still consistently outperforms all the other optimizers for all $v$. We believe LODO is unaffected by noise magnitude because the meta-optimizer Adam has long term

Table 1: Average tracking error of the quadratic bowl minimum on the noisy quadratic bowl task of Section 5.1. Values are averaged over the last 10% of training before the stated training milestone. The theoretically best possible loss using Newton's method is also listed. The version of L-BFGS is one with stochastic modifications from (Schraudolph et al., 2007), instead of the original from (Nocedal & Wright, 1999). Out of the optimizers with reasonable time complexity for scaling to large neural networks (ie. $O(n \log n)$ or less,) LODO performs the best.

| Time complexity | Optimizer | Training loss | |
| --- | --- | --- | --- |
| | | 100k steps | 300 sec. |
| | Adam (Kingma & Ba, 2014) | $15.28 \pm 0.07$ | $15.26 \pm 0.08$ |
| | Momentum | $16.59 \pm 0.08$ | $16.56 \pm 0.05$ |
| | RMSprop (Hinton et al., 2012) | $22.37 \pm 0.06$ | $22.47 \pm 0.13$ |
| $O(n)$ | Yogi (Zaheer et al., 2018) | $15.35 \pm 0.10$ | $15.16 \pm 0.05$ |
| | L-BFGS (Schraudolph et al., 2007) | $49.30 \pm 1.04$ | $40.60 \pm 1.40$ |
| | O-LBFGS (Schraudolph et al., 2007) | $13.96 \pm 0.08$ | $10.80 \pm 0.17$ |
| | L2LBGDBGD (Andrychowicz et al., 2016) | $14.79 \pm 0.08$ | $14.81 \pm 0.10$ |
| $O(n \log n)$ | **LODO (ours)** | $\mathbf{8.99 \pm 0.05}$ | $\mathbf{10.05 \pm 0.22}$ |
| | BFGS (Broyden, 1970) | diverged | diverged |
| $\geq O(n^2)$ | Levenberg-Marquardt (Levenberg, 1944) | $7.40 \pm 0.02$ | $7.35 \pm 0.07$ |
| | Newton Method (Optimal) | 7.41 | |

behavior invariant to gradient magnitudes (ignoring hyperparameter $\epsilon$), meaning LODO's step size behavior is equivariant to gradient magnitudes. This pairs well with the scaling properties of the noisy quadratic bowl problem, which then allow LODO to achieve identical noise-rescaled training loss regardless of noise magnitude.

**Pretrained and Frozen Preconditioner.** To test the importance of LODO learning $\boldsymbol{G}(\boldsymbol{\theta}_t)$ on the fly, we froze $\boldsymbol{G}(\boldsymbol{\theta}_t)$ after using LODO for 100k steps as before and then tried to use this "pre-trained" LODO on a new quadratic bowl. Within 10 steps, the loss diverged to $4.997 \times 10^{22}$, indicating that LODO is best trained on the fly rather than beforehand on a task distribution.

## 5.2 Rosenbrock Function

We probe the behavior of LODO with a small test task of finding the minimum of a rescaled Rosenbrock function $f(x, y) = 0.01(x - 1)^2 + (x^2 - y)^2$, which has no local minima and one global minimum at $(x, y) = (1, 1)$. We initialized the optimizers at $(x, y) = (-0.5, 2)$ and gave them 200 steps to run. The trajectory taken by LODO, shown in Figure 3, is similar to the short timescale dynamics of other optimizers using momentum, in that it tends to overshoot and then correct itself in an oscillatory manner, due to its initialization with the momentum optimization strategy. Learning curves in Figure 11 and losses in Table 8 of Appendix G show the performance of all the optimizers on this task.

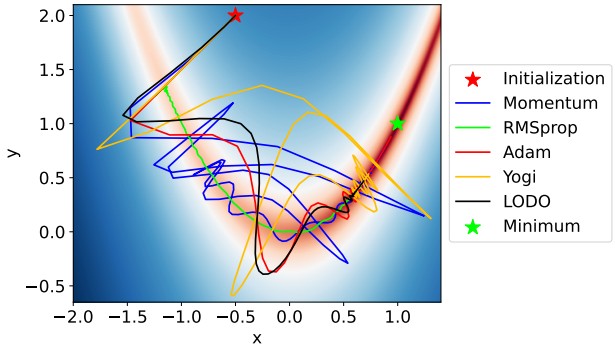

Figure 3: 100 step trajectories of various optimizers on the Rosenbrock function minimization task of Section 5.2. The red star marks the initialization and the green star marks the location of the global minimum.

## 5.3 Image Generation

We design a challenge for LODO—an autoregressive image generator for MNIST. We intend to demonstrate the effectiveness of LODO at scale when used to train a semi-realistic neural network with lots of parameter

sharing. The task is similar to training a PixelCNN (Oord et al., 2016) to generate MNIST images (Lecun et al., 1998), and is fully described in Appendix F.2. We tested LODO alongside a variety optimizers for 300k steps, though we could not get any quasi-Newton methods to converge on this task. We do not compare against K-FAC (Martens & Grosse, 2015b) because this would require matrix inversions of time complexity much larger than the number of CNN weights. Learning curves in Figures 4 and losses in Table 2 show that LODO trains at a speed that is competitive against other optimizers. Figure 10 in Appendix F.2 provides some imitation MNIST images randomly sampled using this model.

Table 2: Negative log likelihoods in nats per pixel after training for 300k steps on the MNIST image generation task of Section 5.3 with every optimizer. Values are averaged over the last 10% of training before the stated training milestone. The top 3 optimizers are underlined for each metric.

| | Training loss | | Test loss | | |
| Optimizer | 300k steps | 50k sec. ($\sim$ 14 h.) | 300k steps | 50k sec. | Steps / sec. |
| --- | --- | --- | --- | --- | --- |
| Adam | $0.830 \pm 0.005$ | $0.859 \pm 0.009$ | 0.809 | 0.854 | $7.08 \pm 0.03$ |
| Momentum | $0.708 \pm 0.005$ | $\underline{0.698 \pm 0.005}$ | $\underline{0.689}$ | $\underline{0.685}$ | $7.10 \pm 0.03$ |
| RMSprop | $0.917 \pm 0.010$ | $0.920 \pm 0.014$ | 0.931 | 0.899 | $7.10 \pm 0.02$ |
| Yogi | $\underline{0.683 \pm 0.002}$ | $\underline{0.677 \pm 0.003}$ | $\underline{0.686}$ | $\underline{0.674}$ | $7.42 \pm 0.02$ |
| LARS (You et al., 2017) | $\underline{0.701 \pm 0.006}$ | $0.702 \pm 0.006$ | $\underline{0.688}$ | $\underline{0.688}$ | $5.89 \pm 0.02$ |
| LODO (ours) | $\underline{0.696 \pm 0.004}$ | $\underline{0.698 \pm 0.005}$ | $\underline{0.689}$ | 0.689 | $5.64 \pm 0.02$ |

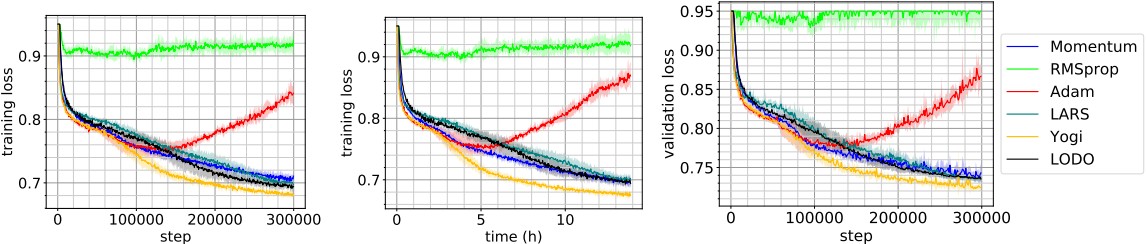

Figure 4: Training loss learning curves on the MNIST image generation task of Section 5.3. **Left:** By step. **Middle:** By time. Our timing setup is described in Appendix H. **Right:** Validation loss by step, using a subset of 64 images excluded from the training data. Each image provides 784 pixel colors to predict, so the validation dataset effectively consists of 50176 samples.

Using this setup, we also study the contributions of various components of LODO to its performance by replacing components to observe a performance change, as detailed below. For example, we replace the Adam meta-optimizer with SGD to form a new version of LODO (which we call "LODO-SGD"); its performance is shown in Table 3. Other modifications are listed below.

### 5.3.1 Residual Connections

We would like to show that there exist opportunities to further develop the architecture of LODO. We modify the matrix decomposition in Section 3 to $\boldsymbol{G}(\boldsymbol{\theta}) = \alpha_0 \begin{pmatrix} \boldsymbol{I} & 0 \end{pmatrix} \tilde{\boldsymbol{G}}(\boldsymbol{\theta})^T \boldsymbol{D} \tilde{\boldsymbol{G}}(\boldsymbol{\theta}) \begin{pmatrix} \boldsymbol{I} & 0 \end{pmatrix}^T$ by adding learned diagonal matrix $\boldsymbol{D}$ in the middle and changing $\tilde{\boldsymbol{G}}(\boldsymbol{\theta})$ to a product of many weighted permutations each added to the identity matrix. The neural network which $\tilde{\boldsymbol{G}}(\boldsymbol{\theta})$ represents now has residual connections, and the initialization is modified accordingly. Losses in Table 3 show that this version (which we call "LODO-Residuals") performs only slightly worse than LODO, reflecting the potential for further development in the architecture design of LODO.

Table 3: Negative log likelihoods in nats per pixel after training for 300k steps on the MNIST image generation task of Section 5.3 with ablated versions of LODO.

| | Training loss | | Test loss | | |
|---|---|---|---|---|---|
| Optimizer | 300k steps | 50k sec. ($\sim$ 14 h.) | 300k steps | 50k sec. | Steps / sec. |
| LODO | $0.696 \pm 0.004$ | $0.698 \pm 0.005$ | 0.689 | 0.689 | $5.64 \pm 0.02$ |
| LODO-Diagonal (Amid et al., 2022) | Diverged | Diverged | Diverged | Diverged | $9.92 \pm 0.09$ |
| LODO-Global (Baydin et al., 2017) | $0.770 \pm 0.035$ | $0.919 \pm 0.139$ | 0.747 | 0.801 | $9.92 \pm 0.03$ |
| LODO-Residuals | $0.701 \pm 0.004$ | $0.750 \pm 0.008$ | 0.693 | 0.741 | $3.31 \pm 0.03$ |
| LODO-No-Momentum | $0.753 \pm 0.007$ | $0.756 \pm 0.007$ | 0.750 | 0.752 | $5.46 \pm 0.06$ |
| LODO-SGD | $0.709 \pm 0.004$ | $0.714 \pm 0.004$ | 0.698 | 0.707 | $5.44 \pm 0.02$ |

### 5.3.2 Simpler Approximate Hessians

We presumed that the representability result of Section 4.3 is only useful because LODO's strength comes from the flexibility that $G(\boldsymbol{\theta})$ gives in configuring pairwise interactions between parameters. We therefore expect that using a simpler form of Hessian should hurt the performance of LODO. We test two simpler forms of $G(\boldsymbol{\theta})$: $G(\boldsymbol{\theta}) = \alpha_0 \text{diag}(\boldsymbol{\theta})$ (which we call "LODO-Diagonal") for $\boldsymbol{\theta} \in \mathbb{R}^n$ initialized to a vector of ones, similar to (Amid et al., 2022)—and the even simpler $G(\boldsymbol{\theta}) = \alpha_0 \theta \boldsymbol{I}$ (which we call "LODO-Global") for $\theta \in \mathbb{R}$ initialized to 1, as in (Baydin et al., 2017). Losses in Table 3 show that the original version of LODO performs the best, verifying our hypothesis.

### 5.3.3 Effects of Using EMAs of Gradients

Similarly to how momentum works for SGD, LODO's input gradients are preproccessed by accumulation into EMAs. To test our claim in Section 3 that momentum benefits LODO, we try 8 separate momentum decay rates spaced in a logarithmic grid from no momentum to the optimal amount of momentum found ($\beta = 0.9343$), and test each decay rate once. Figure 5 shows a clear trend that at least up to the optimal decay rate, increasing the effect of momentum improves LODO. We also try removing momentum completely (we call the modified version "LODO-No-Momentum"); results are shown in Table 3.

### 5.4 Image Classification

We conduct an experiment on image classification with Resnet-18 (He et al., 2016) on CIFAR10 (Krizhevsky et al., 2009). We use the standard Resnet setup on CIFAR10 by replacing the 7x7 convolution with a 3x3 one, and removing the maxpool in the first convolutional block. We use the standard data augmentation and a batch size of 2048.

In Figure 6 and Table 4 we present a more detailed view of the performance of the models, demonstrating the results of our experiment. We observe that LODO performs competitively with the best-performing optimizers (Adam, Yogi, Momentum, and LARS), and noticeably outperforms RMSProp. The "$\geq O(n^2)$" family of optimizers cannot be run on tasks of this scale. Since there are many crucial regularization techniques in computer vision, such as data augmentation and weight decay, a more in-depth study of how LODO interfaces with them is a fruitful direction for future work.

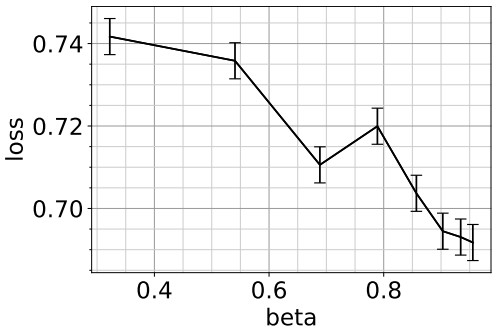

Figure 5: LODO's training loss as a function of the momentum decay coefficient $\beta$, averaged over the last 10% of 300k steps, for the image generation task of Section 5.3. Momentum improves LODO. Error bars depict LODO's uncertainty from Table 2.

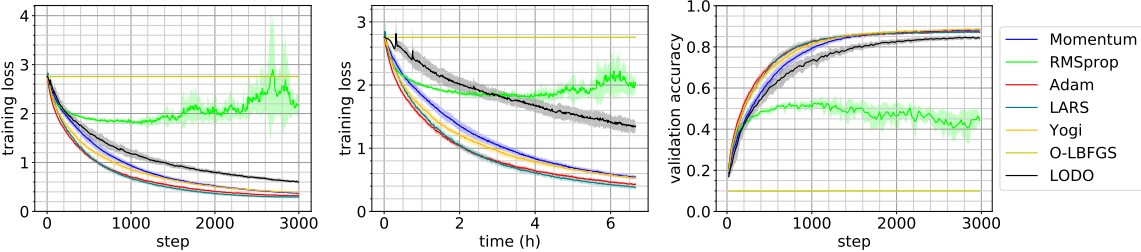

Figure 6: Training loss learning curves on the Resnet-18 CIFAR10 task of Section 5.4. **Left:** By step. **Middle:** By time. Our timing setup is described in Appendix H. **Right:** Validation accuracy by step.

Table 4: Training losses and test accuracies after training a Resnet-18 for 3000 steps (122.88 epochs) or 24k seconds (6.7 hours) on CIFAR10 classification task of Section 5.4 with every optimizer. Values are averaged over the last 10% of training before the stated training milestone.

| Optimizer | Training loss | | Test accuracy | | |
| --- | --- | --- | --- | --- | --- |
| | 3000 steps | 24k sec. ($\sim$ 6.7 h.) | 3000 steps | 24k sec. | Steps / sec. |
| Adam | $0.326 \pm 0.008$ | $0.446 \pm 0.016$ | $0.873 \pm 0.002$ | $0.863 \pm 0.003$ | $0.0793 \pm 0.0003$ |
| Momentum | $0.386 \pm 0.010$ | $0.563 \pm 0.031$ | $0.881 \pm 0.006$ | $0.864 \pm 0.008$ | $0.0794 \pm 0.0002$ |
| RMSprop | $2.380 \pm 0.375$ | $2.077 \pm 0.191$ | $0.434 \pm 0.043$ | $0.466 \pm 0.040$ | $0.0794 \pm 0.0001$ |
| Yogi | $0.398 \pm 0.006$ | $0.552 \pm 0.022$ | $0.885 \pm 0.003$ | $0.869 \pm 0.005$ | $0.0791 \pm 0.0004$ |
| LARS (You et al., 2017) | $0.296 \pm 0.026$ | $0.403 \pm 0.034$ | $0.872 \pm 0.006$ | $0.864 \pm 0.007$ | $0.0788 \pm 0.0003$ |
| LODO (ours) | $0.624 \pm 0.027$ | $1.377 \pm 0.110$ | $0.845 \pm 0.009$ | $0.674 \pm 0.035$ | $0.0316 \pm 0.0003$ |

## 6 Discussion

LODO is a cross between L2O methods and quasi-Newton methods, retaining significant advantages of both classes of optimizers. With LODO, we bring ideas from each class of optimization methods to the other.

Relative to quasi-Newton methods, LODO offers advantages associated with the use of a meta-optimizer on a neural optimizer. Crucially, LODO determines its inverse Hessian estimate using all past gradients, whereas most other quasi-Newton methods use a finite history of them. This allows LODO to retain information about the inverse Hessian for much longer than other methods. This is useful if the gradients contain enough noise that useful signals can only be obtained by accumulating information from many gradient samples. Our theory further shows that the sparse linear neural network in LODO is optimal to a certain extent: it can probably represent all sparse linear neural networks smaller by a logarithmic factor—allowing for a huge class of inverse Hessians. Our image generation task demonstrates that LODO succeeds in a semi-realistic stochastic nonconvex task where other quasi-Newton optimizers diverge. Due to our use of L2O, LODO can be further developed in the design of its linear neural network, which makes it amenable to further research and refinement.

Relative to L2O, LODO offers advantages associated with the restructuring of the outer and inner loop into a single loop. Most importantly, our modification to L2O alleviates the requirement for meta-training time and the training task distribution. This is at the cost of increased inner loop unrolling truncation bias, but it takes advantage of this sacrifice by resolving the need to compute second-order gradients. LODO still inherits issues of high memory usage and slow step computation from L2O methodology though. Our theory offers some understanding of how LODO learns to optimize: the Hessian approximation error decays as learning progresses. We import the idea from quasi-Newton methods that the gradient of one parameter can affect the step for another, which comes from the presence of off-diagonal elements in the Hessian. As shown in Section 4.3, LODO presents an efficient way of approximating subsets of the $O(n^2)$ possible pairwise parameter interactions in $O(n \log n)$ time. Such interactions are completely ignored in the design of most

L2O and more mainstream optimizers, yet our image generation task demonstrates their importance, as evidenced by the improved performance of LODO over ablated versions as well as SGD.

## 7 Conclusion

Through LODO, we provide a new way of using L2O methods online without any meta-training to perform quasi-Newton optimization. We introduce the strengths and advantages of quasi-Newton methods and L2O to each other and combine them in a harmonious manner. LODO's abilities showcase the applicability of online L2O methods with nested optimizers to the training of modern neural networks. Our unique methodology serves as a stepping stone for the further development and use of L2O in quasi-Newton optimization and vice versa.

## 8 Acknowledgements

We would like to acknowledge the MIT SuperCloud and Lincoln Laboratory Supercomputing Center for providing HPC resources that have contributed to the research results reported within this paper.

Research was sponsored by the United States Air Force Research Laboratory and the United States Air Force Artificial Intelligence Accelerator and was accomplished under Cooperative Agreement Number FA8750-19-2-1000. The views and conclusions contained in this document are those of the authors and should not be interpreted as representing the official policies, either expressed or implied, of the United States Air Force or the U.S. Government. The U.S. Government is authorized to reproduce and distribute reprints for Government purposes notwithstanding any copyright notation herein.

This work was also sponsored in part by the the National Science Foundation under Cooperative Agreement PHY-2019786 (The NSF AI Institute for Artificial Intelligence and Fundamental Interactions, `http://iaifi.org/`).

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

## A Elaboration on Hessian Learning Dynamics

### A.1 Derivation of Training Dynamics

This section gives a derivation of the result that under the problem setup of Section 4.1, LODO follows the Hessian learning dynamics

$$\boldsymbol{A}_{t+1} = \boldsymbol{A}_t - \alpha \boldsymbol{H} \boldsymbol{b}_{t+1} \boldsymbol{b}_t^T \boldsymbol{H}^2 \tag{5}$$

$$\boldsymbol{b}_{t+1} = \boldsymbol{A}_t \boldsymbol{b}_t - \boldsymbol{s}_t, \tag{6}$$

where $\boldsymbol{A}_t = \boldsymbol{I} - \boldsymbol{G}(\boldsymbol{\theta}_t) \boldsymbol{H}$ and $\boldsymbol{b}_t = \boldsymbol{x}_t - \boldsymbol{x}_t^*$ as long as $\boldsymbol{G}(\boldsymbol{\theta}_t)$ is parameterized as a dense matrix filled with elements of $\boldsymbol{\theta}_t$, and no momentum is used.

We first let $\boldsymbol{b}_t$ be $\boldsymbol{x}_t - \boldsymbol{x}_t^*$. Following Equation 3, the loss at time $t$ is then

$$\ell_t = \frac{1}{2} \boldsymbol{b}_t^T \boldsymbol{H} \boldsymbol{b}_t. \tag{17}$$

The gradient is then computed to be

$$\frac{\mathrm{d}\ell}{\mathrm{d}\boldsymbol{x}_t} = \boldsymbol{H} \boldsymbol{b}_t. \tag{18}$$

The step taken then produces the next parameters:

$$\boldsymbol{x}_{t+1} = \boldsymbol{x}_t - \boldsymbol{G}(\boldsymbol{\theta}_t) \boldsymbol{H} \boldsymbol{b}_t. \tag{19}$$

Subtracting $\boldsymbol{x}_{t+1}^* = \boldsymbol{x}_t^* + \boldsymbol{s}_t$, we get the recurrence for $\boldsymbol{b}_t$,

$$\boldsymbol{x}_{t+1} - \boldsymbol{x}_{t+1}^* = \boldsymbol{x}_t - \boldsymbol{x}_t^* - \boldsymbol{s}_t - \boldsymbol{G}(\boldsymbol{\theta}_t) \boldsymbol{H} \boldsymbol{b}_t \tag{20}$$

$$\boldsymbol{b}_{t+1} = \boldsymbol{b}_t - \boldsymbol{G}(\boldsymbol{\theta}_t) \boldsymbol{H} \boldsymbol{b}_t - \boldsymbol{s}_t \tag{21}$$

$$= (\boldsymbol{I} - \boldsymbol{G}(\boldsymbol{\theta}_t) \boldsymbol{H}) \boldsymbol{b}_t - \boldsymbol{s}_t \tag{22}$$

$$= \boldsymbol{A}_t \boldsymbol{b}_t - \boldsymbol{s}_t. \tag{6}$$

The loss at time $t + 1$ is computed to be

$$\ell_{t+1} = \frac{1}{2} \boldsymbol{b}_{t+1}^T \boldsymbol{H} \boldsymbol{b}_{t+1} \tag{23}$$

$$= \frac{1}{2} (\boldsymbol{A}_t \boldsymbol{b}_t - \boldsymbol{s}_t)^T \boldsymbol{H} (\boldsymbol{A}_t \boldsymbol{b}_t - \boldsymbol{s}_t) \tag{24}$$

$$= \frac{1}{2} ((\boldsymbol{I} - \boldsymbol{G}(\boldsymbol{\theta}_t) \boldsymbol{H}) \boldsymbol{b}_t - \boldsymbol{s}_t)^T \boldsymbol{H} ((\boldsymbol{I} - \boldsymbol{G}(\boldsymbol{\theta}_t) \boldsymbol{H}) \boldsymbol{b}_t - \boldsymbol{s}_t). \tag{25}$$

LODO also computes a step of $\boldsymbol{\theta}_t$ using the loss on the next step. Since the elements of $\boldsymbol{\theta}_t$ are just a rearrangement of the elements of $\boldsymbol{G}(\boldsymbol{\theta}_t)$ in our derivation, an update of $\boldsymbol{\theta}_t$ can be treated instead like an update of $\boldsymbol{G}(\boldsymbol{\theta}_t)$. The gradient of $\ell_{t+1}$ with respect to $\boldsymbol{G}(\boldsymbol{\theta}_t)$ is then computed to be

$$\frac{\mathrm{d}\ell_{t+1}}{\mathrm{d}\boldsymbol{G}(\boldsymbol{\theta}_t)} = - \boldsymbol{H} ((\boldsymbol{I} - \boldsymbol{G}(\boldsymbol{\theta}_t) \boldsymbol{H}) \boldsymbol{b}_t - \boldsymbol{s}_t) \boldsymbol{b}_t^T \boldsymbol{H} \tag{26}$$

$$= - \boldsymbol{H} (\boldsymbol{A}_t \boldsymbol{b}_t - \boldsymbol{s}_t) \boldsymbol{b}_t^T \boldsymbol{H} \tag{27}$$

$$= - \boldsymbol{H} \boldsymbol{b}_{t+1} \boldsymbol{b}_t^T \boldsymbol{H} \tag{28}$$

and the step of $\boldsymbol{G}(\boldsymbol{\theta}_t)$ is

$$\boldsymbol{G}(\boldsymbol{\theta}_{t+1}) = \boldsymbol{G}(\boldsymbol{\theta}_t) + \alpha \boldsymbol{H}\boldsymbol{b}_{t+1}\boldsymbol{b}_t^T\boldsymbol{H} \tag{29}$$

resulting in the recurrence for $\boldsymbol{A}_t$:

$$\boldsymbol{A}_{t+1} = \boldsymbol{I} - \boldsymbol{G}(\boldsymbol{\theta}_{t+1})\boldsymbol{H} \tag{30}$$
$$= \boldsymbol{I} - (\boldsymbol{G}(\boldsymbol{\theta}_t) + \alpha \boldsymbol{H}\boldsymbol{b}_{t+1}\boldsymbol{b}_t^T\boldsymbol{H})\boldsymbol{H} \tag{31}$$
$$= \boldsymbol{A}_t - \alpha \boldsymbol{H}\boldsymbol{b}_{t+1}\boldsymbol{b}_t^T\boldsymbol{H}^2. \tag{5}$$

## A.2 Validity of Approximation Argument

This section gives justification for the approximation in Section 4.1 of the long term trajectory of the recurrence

$$\boldsymbol{A}_{t+1} = \boldsymbol{A}_t - \alpha \boldsymbol{H}\boldsymbol{b}_{t+1}\boldsymbol{b}_t^T\boldsymbol{H}^2 \tag{5}$$
$$\boldsymbol{b}_{t+1} = \boldsymbol{A}_t\boldsymbol{b}_t - \boldsymbol{s}_t \tag{6}$$

by replacing with

$$\boldsymbol{A}'_{t+1} = \boldsymbol{A}'_t - \alpha \boldsymbol{H}\boldsymbol{b}'_{t+1}\boldsymbol{b}'^T_t\boldsymbol{H}^2 \tag{32}$$
$$\boldsymbol{b}'_{t+1} = \boldsymbol{A}'_{t_0}\boldsymbol{b}'_t - \boldsymbol{s}_t \tag{33}$$

when $\alpha$ is small and the initial conditions at $t_0$ are the same: $\boldsymbol{A}_{t_0} = \boldsymbol{A}'_{t_0}$ and $\boldsymbol{b}_{t_0} = \boldsymbol{b}'_{t_0} = 0$. We will work in the bounded noise case $||\boldsymbol{s}_t||_2 < \infty$, where $||\boldsymbol{b}'_t||_2$ is upper bounded by some $||\boldsymbol{A}'_{t_0}||_2$ dependent constant $b_{\max}$ due to exponential decay in Equation 33. In the case where the noise is not bounded, a probabilistic analysis can be done instead, though we do not provide one.

To justify this approximation, we prove that the spectral norm of long term deviation corrected for learning rate is small over short distances $r$, in the following theorem:

**Theorem A.1.**

$$\lim_{r \to 0} \lim_{\alpha \to 0} \frac{1}{r}||\boldsymbol{A}_{t_0+\lfloor r/\alpha \rfloor} - \boldsymbol{A}'_{t_0+\lfloor r/\alpha \rfloor}||_2 = 0. \tag{34}$$

In other words, the local movement of $\boldsymbol{A}$ rescaled for learning rate is unaffected by our approximation when the learning rate $\alpha$ is small.

*Proof.* Our proof strategy is as follows:

1. We will first define variables to denote bounds on the movement of $\boldsymbol{A}$ and the approximation error in $\boldsymbol{b}$.

2. We will show that these variables bound each other, and then we will combine these bounds to create a single recursive bound on the movement of $\boldsymbol{A}$.

3. We will characterize the bound's growth and it will turn out that $\boldsymbol{A}$ has a maximum movement speed along any trajectory of sufficiently short length.

4. Due to the slow movement of $\boldsymbol{A}$, we can deduce that the approximation error in $\boldsymbol{b}$ increases at a bounded rate.

5. Since approximation errors in $\boldsymbol{A}$ are an accumulation of errors in $\boldsymbol{b}$, we will show that deviation between the true and approximate $\boldsymbol{A}$ trajectories is quadratic in the distance along the trajectory.

6. We conclude that the approximation error vanishes for short trajectories and small learning rates.

**First part.** We first define the maximum drift in $\boldsymbol{A}$

$$\epsilon_{\boldsymbol{A},t_0+\Delta t} = \max_{t_0 \leq \tau \leq t_0+\Delta t} ||\boldsymbol{A}_\tau - \boldsymbol{A}'_{t_0}||_2 \tag{35}$$

up to time difference $\Delta t$ for $0 \leq \Delta t \leq R/\alpha$ for some chosen small constant trajectory length $R > 0$. We will pick $R$ later. We will also define the maximum error in $\boldsymbol{b}$

$$\epsilon_{\boldsymbol{b},t_0+\Delta t} = \max_{t_0 \leq \tau \leq t_0+\Delta t} ||\boldsymbol{b}_\tau - \boldsymbol{b}'_\tau||_2 \tag{36}$$

up to the same time.

**Second part.** For the bound in one direction, we have that for all $\tau$ such that $t_0 \leq \tau \leq t_0 + \Delta t$,

$$||\boldsymbol{b}_{\tau+1} - \boldsymbol{b}'_{\tau+1}||_2 = ||\boldsymbol{A}_\tau \boldsymbol{b}_\tau - \boldsymbol{s}_\tau - (\boldsymbol{A}'_{t_0}\boldsymbol{b}'_\tau - \boldsymbol{s}_\tau)||_2 \tag{37}$$

$$= ||\boldsymbol{A}_\tau \boldsymbol{b}_\tau - \boldsymbol{A}'_{t_0}\boldsymbol{b}'_\tau||_2 \tag{38}$$

$$\leq ||\boldsymbol{A}_\tau \boldsymbol{b}_\tau - \boldsymbol{A}'_{t_0}\boldsymbol{b}_\tau||_2 + ||\boldsymbol{A}'_{t_0}\boldsymbol{b}_\tau - \boldsymbol{A}'_{t_0}\boldsymbol{b}'_\tau||_2 \tag{39}$$

$$\leq ||\boldsymbol{A}_\tau - \boldsymbol{A}'_{t_0}||_2||\boldsymbol{b}_\tau||_2 + ||\boldsymbol{A}'_{t_0}||_2||\boldsymbol{b}_\tau - \boldsymbol{b}'_\tau||_2 \tag{40}$$

$$\leq \epsilon_{\boldsymbol{A},t_0+\Delta t}||\boldsymbol{b}_\tau||_2 + ||\boldsymbol{A}'_{t_0}||_2||\boldsymbol{b}_\tau - \boldsymbol{b}'_\tau||_2 \tag{41}$$

using the triangle inequality and sub-multiplicativity for the spectral norm $||\cdot||_2$. This is a recurrence in $||\boldsymbol{b}_\tau - \boldsymbol{b}'_\tau||_2$; by induction we have that for $t_0 \leq \tau \leq t_0 + \Delta t + 1$,

$$||\boldsymbol{b}_\tau - \boldsymbol{b}'_\tau||_2 \leq \epsilon_{\boldsymbol{A},t_0+\Delta t} \sum_{\tau_1=t_0}^{\tau-1} ||\boldsymbol{A}'_{t_0}||_2^{\tau-1-\tau_1}||\boldsymbol{b}_{\tau_1}||_2 \tag{42}$$

such that we produce the bound

$$\epsilon_{\boldsymbol{b},t_0+\Delta t+1} \leq \epsilon_{\boldsymbol{A},t_0+\Delta t} \max_{t_0 \leq \tau \leq t_0+\Delta t+1} \sum_{\tau_1=t_0}^{\tau-1} ||\boldsymbol{A}'_{t_0}||_2^{\tau-1-\tau_1}||\boldsymbol{b}_{\tau_1}||_2 \tag{43}$$

$$\leq \epsilon_{\boldsymbol{A},t_0+\Delta t} \max_{t_0 \leq \tau \leq t_0+\Delta t+1} \sum_{\tau_1=t_0}^{\tau-1} ||\boldsymbol{A}'_{t_0}||_2^{\tau-1-\tau_1}(||\boldsymbol{b}'_{\tau_1}||_2 + \epsilon_{\boldsymbol{b},t_0+\Delta t}) \tag{44}$$

$$\leq \epsilon_{\boldsymbol{A},t_0+\Delta t} \sum_{\tau_1=t_0}^{t_0+\Delta t} ||\boldsymbol{A}'_{t_0}||_2^{t_0+\Delta t-\tau_1}(b_{\max} + \epsilon_{\boldsymbol{b},t_0+\Delta t}) \tag{45}$$

$$\leq \epsilon_{\boldsymbol{A},t_0+\Delta t}\frac{\epsilon_{\boldsymbol{b},t_0+\Delta t+1} + b_{\max}}{1 - ||\boldsymbol{A}'_{t_0}||_2} \tag{46}$$

$$\epsilon_{\boldsymbol{b},t_0+\Delta t+1} \leq \frac{\epsilon_{\boldsymbol{A},t_0+\Delta t}b_{\max}}{1 - ||\boldsymbol{A}'_{t_0}||_2 - \epsilon_{\boldsymbol{A},t_0+\Delta t}}. \tag{47}$$

Now, we show a reverse bound: for all $\tau$ such that $t_0 \leq \tau \leq t_0 + \Delta t$, we have

$$||\boldsymbol{A}_{\tau+1} - \boldsymbol{A}'_{t_0}||_2 = ||\boldsymbol{A}_\tau - \alpha \boldsymbol{H}\boldsymbol{b}_{\tau+1}\boldsymbol{b}_\tau^T \boldsymbol{H}^2 - \boldsymbol{A}'_{t_0}||_2 \tag{48}$$

$$\leq ||\boldsymbol{A}_\tau - \boldsymbol{A}'_{t_0}||_2 + \alpha||\boldsymbol{H}||_2^3||\boldsymbol{b}_\tau||_2||\boldsymbol{b}_{\tau+1}||_2 \tag{49}$$

$$\leq ||\boldsymbol{A}_\tau - \boldsymbol{A}'_{t_0}||_2 + \alpha||\boldsymbol{H}||_2^3(||\boldsymbol{b}'_\tau||_2 + \epsilon_{\boldsymbol{b},t_0+\Delta t+1})(||\boldsymbol{b}'_{\tau+1}||_2 + \epsilon_{\boldsymbol{b},t_0+\Delta t+1}) \tag{50}$$

$$\leq ||\boldsymbol{A}_\tau - \boldsymbol{A}'_{t_0}||_2 + \alpha||\boldsymbol{H}||_2^3(b_{\max} + \epsilon_{\boldsymbol{b},t_0+\Delta t+1})^2 \tag{51}$$

By induction we have for $t_0 \leq \tau \leq t_0 + \Delta t + 1$,

$$||\boldsymbol{A}_\tau - \boldsymbol{A}'_{t_0}||_2 \leq \alpha||\boldsymbol{H}||_2^3(\tau - t_0)(b_{\max} + \epsilon_{\boldsymbol{b},t_0+\Delta t+1})^2 \tag{52}$$

such that we produce the reverse bound

$$\epsilon_{\boldsymbol{A},t_0+\Delta t+1} \leq \alpha||\boldsymbol{H}||_2^3(\Delta t + 1)(b_{\max} + \epsilon_{\boldsymbol{b},t_0+\Delta t+1})^2. \tag{53}$$

**Third part.** Substituting the bound in Equation 47 into the bound in Equation 53, we produce the recurrence

$$\epsilon_{\boldsymbol{A},t_0+\Delta t+1} \leq \alpha ||\boldsymbol{H}||_2^3 b_{\max}^2 (\Delta t + 1) \left( 1 + \frac{\epsilon_{\boldsymbol{A},t_0+\Delta t}}{1 - ||\boldsymbol{A}'_{t_0}||_2 - \epsilon_{\boldsymbol{A},t_0+\Delta t}} \right)^2 \tag{54}$$

$$= \alpha ||\boldsymbol{H}||_2^3 b_{\max}^2 (\Delta t + 1) \left( \frac{1 - ||\boldsymbol{A}'_{t_0}||_2}{1 - ||\boldsymbol{A}'_{t_0}||_2 - \epsilon_{\boldsymbol{A},t_0+\Delta t}} \right)^2 \tag{55}$$

$$= f(\epsilon_{\boldsymbol{A},t_0+\Delta t}). \tag{56}$$

where

$$f(x) = \alpha ||\boldsymbol{H}||_2^3 b_{\max}^2 (\Delta t + 1) \left( \frac{1 - ||\boldsymbol{A}'_{t_0}||_2}{1 - ||\boldsymbol{A}'_{t_0}||_2 - x} \right)^2. \tag{57}$$

To bound the movement of $\boldsymbol{A}$, we must use the fact that when

$$0 \leq \Delta t \leq \frac{4}{27} \frac{1 - ||\boldsymbol{A}'_{t_0}||_2}{\alpha ||\boldsymbol{H}||_2^3 b_{\max}^2} - 1 \tag{58}$$

the function $f$ maps the interval

$$I_{\Delta t} = \left[ 0, \frac{9}{4} \alpha ||\boldsymbol{H}||_2^3 b_{\max}^2 (\Delta t + 1) \right] \subseteq \left[ 0, \frac{1}{3}(1 - ||\boldsymbol{A}'_{t_0}||_2) \right] \tag{59}$$

to a subset of itself. Since at $\Delta t = 0$ we have $\epsilon_{\boldsymbol{A},t_0+\Delta t} = 0 \in I_{\Delta t}$, and we also have $I_{\Delta t} \subseteq I_{\Delta t+1}$, we may deduce by induction on $\Delta t$ that $\epsilon_{\boldsymbol{A},t_0+\Delta t} \in I_{\Delta t}$ as long as Equation 58 holds, and thus there is a bound

$$\epsilon_{\boldsymbol{A},t_0+\Delta t} \leq \frac{9}{4} \alpha ||\boldsymbol{H}||_2^3 b_{\max}^2 (\Delta t + 1) \leq \frac{1}{3}(1 - ||\boldsymbol{A}'_{t_0}||_2) \tag{60}$$

on the movement speed of $\boldsymbol{A}$ as long as Equation 58 holds.

**Fourth part.** Note that we have assumed that $0 \leq \Delta t \leq R/\alpha$ for some constant $R$ which we have not yet picked. By choosing

$$R \leq \frac{4}{27} \frac{1 - ||\boldsymbol{A}'_{t_0}||_2}{||\boldsymbol{H}||_2^3 b_{\max}^2} - \alpha \tag{61}$$

we may always guarantee Equation 58, which implies Equation 60. Then when Equation 60 is substituted into Equation 47, we create a small bound on the approximation error in $\boldsymbol{b}$ which begins at zero and increases with time,

$$\epsilon_{\boldsymbol{b},t_0+\Delta t+1} \leq \frac{\frac{9}{4}\alpha ||\boldsymbol{H}||_2^3 b_{\max}^3 (\Delta t + 1)}{1 - ||\boldsymbol{A}'_{t_0}||_2 - \frac{9}{4}\alpha ||\boldsymbol{H}||_2^3 b_{\max}^2 (\Delta t + 1)} \tag{62}$$

$$\leq \frac{\alpha b_{\max}(\Delta t + 1)}{3R - \alpha(\Delta t + 1)} \tag{63}$$

for $0 \leq \Delta t \leq R/\alpha$. This also holds trivially for $\Delta t = -1 \implies \epsilon_{\boldsymbol{b},t_0+\Delta t+1} = 0$, so we may re-index to have

$$\epsilon_{\boldsymbol{b},t_0+\Delta t} \leq \frac{\alpha b_{\max}\Delta t}{3R - \alpha\Delta t} \tag{64}$$

for $0 \leq \Delta t \leq R/\alpha + 1$. Since the right side of Equation 64 is convex in $\Delta t$ over $\Delta t \in [0, R/\alpha]$, we may bound by a linear function with the same endpoints

$$\epsilon_{\boldsymbol{b},t_0+\Delta t} \leq \frac{\alpha b_{\max}}{2R} \Delta t \tag{65}$$

for $0 \leq \Delta t \leq R/\alpha$.

**Fifth part.** Finally, we use this bound on approximation error in $\boldsymbol{b}$ to bound approximation error in $\boldsymbol{A}$.

$$||\boldsymbol{A}_{t_0+\Delta t+1} - \boldsymbol{A}'_{t_0+\Delta t+1}||_2$$

$$=||\boldsymbol{A}_{t_0+\Delta t} - \alpha \boldsymbol{H}\boldsymbol{b}_{t_0+\Delta t+1}\boldsymbol{b}_{t_0+\Delta t}^T\boldsymbol{H}^2 - (\boldsymbol{A}'_{t_0+\Delta t} - \alpha \boldsymbol{H}\boldsymbol{b}'_{t_0+\Delta t+1}\boldsymbol{b}'^{T}_{t_0+\Delta t}\boldsymbol{H}^2)||_2 \tag{66}$$

$$\leq ||\boldsymbol{A}_{t_0+\Delta t} - \boldsymbol{A}'_{t_0+\Delta t}||_2 + \alpha ||\boldsymbol{H}||^3 ||\boldsymbol{b}_{t_0+\Delta t+1}\boldsymbol{b}_{t_0+\Delta t}^T - \boldsymbol{b}'_{t_0+\Delta t+1}\boldsymbol{b}'^{T}_{t_0+\Delta t}||_2 \tag{67}$$

$$\leq ||\boldsymbol{A}_{t_0+\Delta t} - \boldsymbol{A}'_{t_0+\Delta t}||_2 + \alpha ||\boldsymbol{H}||^3 \bigg( ||\boldsymbol{b}_{t_0+\Delta t+1}||_2 ||\boldsymbol{b}_{t_0+\Delta t}^T - \boldsymbol{b}'^{T}_{t_0+\Delta t}||_2$$

$$+ ||\boldsymbol{b}_{t_0+\Delta t+1} - \boldsymbol{b}'_{t_0+\Delta t+1}||_2 ||\boldsymbol{b}'^{T}_{t_0+\Delta t}||_2 \bigg) \tag{68}$$

$$\leq ||\boldsymbol{A}_{t_0+\Delta t} - \boldsymbol{A}'_{t_0+\Delta t}||_2 + \epsilon_{\boldsymbol{b},t_0+\Delta t+1}\alpha ||\boldsymbol{H}||^3 (2b_{\max} + \epsilon_{\boldsymbol{b},t_0+\Delta t+1}) \tag{69}$$

By induction, we find that the approximation error of $\boldsymbol{A}$ is quadratic in time for short times $0 \leq \Delta t \leq R/\alpha$,

$$||\boldsymbol{A}_{t_0+\Delta t} - \boldsymbol{A}'_{t_0+\Delta t}||_2 \leq \sum_{\widetilde{\Delta t}=0}^{\Delta t-1} \epsilon_{\boldsymbol{b},t_0+\widetilde{\Delta t}+1}\alpha ||\boldsymbol{H}||^3 (2b_{\max} + \epsilon_{\boldsymbol{b},t_0+\widetilde{\Delta t}+1}) \tag{70}$$

$$=||\boldsymbol{H}||^3 b_{\max}^2 \sum_{\widetilde{\Delta t}=0}^{\Delta t-1} \frac{\alpha^2}{2R}(\widetilde{\Delta t}+1)\left(2 + \frac{\alpha}{2R}(\widetilde{\Delta t}+1)\right) \tag{71}$$

$$\leq ||\boldsymbol{H}||^3 b_{\max}^2 \sum_{\widetilde{\Delta t}=1}^{\Delta t} \frac{\alpha^2}{2R}\Delta t \left(2 + \frac{\alpha}{2R}\Delta t\right) \tag{72}$$

$$=||\boldsymbol{H}||^3 b_{\max}^2 \frac{\alpha^2 \Delta t^2}{2R}\left(2 + \frac{\alpha \Delta t}{2R}\right). \tag{73}$$

**Sixth part.** Now take $\Delta t = \lfloor r/\alpha \rfloor$, which for $r \to 0$ is eventually $\leq R/\alpha$ as required. As we sought to prove, the learning rate rescaled approximation error of the local drift direction and speed goes to zero:

$$\lim_{r \to 0}\lim_{\alpha \to 0} \frac{1}{r}||\boldsymbol{A}_{t_0+\lfloor r/\alpha \rfloor} - \boldsymbol{A}'_{t_0+\lfloor r/\alpha \rfloor}||_2 = \lim_{r \to 0}\lim_{\alpha \to 0}\frac{1}{r}||\boldsymbol{H}||^3 b_{\max}^2 \frac{\alpha^2 \lfloor r/\alpha \rfloor^2}{2R}\left(2 + \frac{\alpha \lfloor r/\alpha \rfloor}{2R}\right) \tag{74}$$

$$= \lim_{r \to 0}||\boldsymbol{H}||^3 b_{\max}^2 \frac{r}{2R}\left(2 + \frac{r}{2R}\right) \tag{75}$$

$$=0. \tag{34}$$

$\square$

## B  Proof that $\mathbf{G}(\theta_t)\mathbf{H}$ has a Negative Eigenvalue

This is true, because we can substitute $\boldsymbol{A} = \mathbf{G}(\theta_t)$ and $\boldsymbol{B} = \mathbf{H}$ in following lemma:

**Lemma B.1.** *Let $\boldsymbol{A}$ and $\boldsymbol{B}$ be symmetric, full-rank $n \times n$ matrices. Let $\boldsymbol{A}$ be positive-definite and $\boldsymbol{B}$ have at least one negative eigenvalue. Then, the product $\boldsymbol{AB}$ has at least one negative eigenvalue.*

*Proof.* Let $\boldsymbol{x}$ be an eigenvector of $\boldsymbol{B}$ with negative eigenvalue $\lambda$. Then, we have

$$\boldsymbol{x}^T\boldsymbol{B}\boldsymbol{x} = \left(\boldsymbol{A}^{-1/2}\boldsymbol{x}\right)^T \boldsymbol{A}^{1/2}\boldsymbol{B}\boldsymbol{A}^{1/2}\left(\boldsymbol{A}^{-1/2}\boldsymbol{x}\right) < 0 \tag{76}$$

meaning that the symmetric matrix $\boldsymbol{A}^{1/2}\boldsymbol{B}\boldsymbol{A}^{1/2}$ cannot possibly be positive-semidefinite, and must have at least one negative eigenvalue $\lambda'$ with eigenvector $\boldsymbol{x}'$. Then, we have the eigenvalue equation

$$\boldsymbol{AB}\left(\boldsymbol{A}^{1/2}\boldsymbol{x}'\right) = \boldsymbol{A}^{1/2}\left(\boldsymbol{A}^{1/2}\boldsymbol{B}\boldsymbol{A}^{1/2}\right)\boldsymbol{x}' \tag{77}$$

$$=\lambda'\boldsymbol{A}^{1/2}\boldsymbol{x}' \tag{78}$$

which shows that $\boldsymbol{AB}$ has a negative eigenvalue $\lambda'$. $\square$

# C   Proof of Representability Theorem

This section gives a proof of the representability theorem stated in Section 4.3:

**Theorem 4.2.** *Uniformly sample permutations $\boldsymbol{P}_i$ and create block-diagonal matrices $\boldsymbol{B}(\boldsymbol{\theta}^{(i)})$ where every block is $2 \times 2$, and whose block contents are listed by the parameters $\boldsymbol{\theta}^{(i)}$. Use these to construct the LODO subnetwork $\tilde{\boldsymbol{G}}(\boldsymbol{\theta})$ as in Equation 2 with some depth $N$ and hidden dimension $\tilde{n}$. Construct any linear neural network $\tilde{\boldsymbol{F}}$ with input dimension, output dimension, number of connections per layer at most $\tilde{n}$, at most $k$ incoming and at most $k$ outgoing connections for every neuron, depth $d$, and otherwise any arrangement of connections. Then, there is a probability of at least*

$$1 - \tilde{n}!N\sqrt{\frac{1}{2}\epsilon\left(\frac{\tilde{n}N}{4d(\lceil \log_2 k \rceil + 1)}, \tilde{n}\right)} \tag{15}$$

*that $\tilde{\boldsymbol{G}}(\boldsymbol{\theta}) = \tilde{\boldsymbol{F}}$ for some $\boldsymbol{\theta}$.*

*Proof.* Our result comes in two parts: the first shows that $\tilde{\boldsymbol{G}}(\boldsymbol{\theta})$ can represent arbitrary permutations, and the second shows that we can pick these $\tilde{\boldsymbol{G}}(\boldsymbol{\theta})$ permutations and interleave them with block diagonal matrices to create a deeper $\tilde{\boldsymbol{G}}(\boldsymbol{\theta})$ network which manifests any desired neural network. We use the terms fanin and fanout to mean the number of incoming and outgoing weights into and out of a neuron, respectively.

**First part.**   Assume we would like to apply a given target permutation to $\tilde{n}$ elements using $\tilde{\boldsymbol{G}}(\boldsymbol{\theta}) = \prod_{i=1}^{N} \boldsymbol{B}(\boldsymbol{\theta}^{(i)})\boldsymbol{P}_i$ consisting of $N$ layers with randomly chosen permutations $\boldsymbol{P}_i$. Our goal is to perform the target permutation given $\boldsymbol{P}_i$ by controlling the block diagonal matrices $\boldsymbol{B}(\boldsymbol{\theta}^{(i)})$. The form of $\tilde{\boldsymbol{G}}(\boldsymbol{\theta})$ from Section 3 is

$$\tilde{\boldsymbol{G}}(\boldsymbol{\theta}) = \prod_{i=1}^{N} \boldsymbol{B}(\boldsymbol{\theta}^{(i)})\boldsymbol{P}_i \tag{2}$$

which can be rewritten as

$$\tilde{\boldsymbol{G}}(\boldsymbol{\theta}) = \boldsymbol{Q}_1 \prod_{i=1}^{N} \boldsymbol{Q}_i^T \boldsymbol{B}(\boldsymbol{\theta}^{(i)})\boldsymbol{Q}_i, \qquad \boldsymbol{Q}_N = \boldsymbol{P}_i, \quad \boldsymbol{Q}_{i-1} = \boldsymbol{P}_{i-1}\boldsymbol{Q}_i \tag{79}$$

with random independent uniform permutations $\boldsymbol{Q}_i$ instead. For each block in the matrix $\boldsymbol{B}(\boldsymbol{\theta}^{(i)})$, we may restrict ourselves to two options: to swap or to not swap the pair of indices. The conjugation by random permutation $\boldsymbol{Q}_i$ then shuffles these pairings, such that applying $\tilde{\boldsymbol{G}}(\boldsymbol{\theta})$ is equivalent to repeatedly randomly pairing up indices for optional transposition instead, and then applying a final permutation $\boldsymbol{Q}_1$. We will work under this new equivalent formulation, since it is more amenable to analysis.

Let us choose to apply each transposition with probability $1/2$. Then, the expected entropy of $\tilde{\boldsymbol{G}}(\boldsymbol{\theta})$ given the whole sequence of pairings is at least $\log \tilde{n}! - \epsilon(N\tilde{n}/2, \tilde{n})$ under Definition 4.1. In other words, the expected KL divergence of this distribution of $\tilde{\boldsymbol{G}}(\boldsymbol{\theta})$ from the uniform is at most $\epsilon(N\tilde{n}/2, \tilde{n})$. Then by Pinsker's inequality, the expected total variation distance from the uniform is then at most

$$\sqrt{\frac{1}{2}\epsilon(N\tilde{n}/2, \tilde{n})}. \tag{80}$$

This guarantees that at most

$$\tilde{n}!\sqrt{\frac{1}{2}\epsilon(N\tilde{n}/2, \tilde{n})} \tag{81}$$

possible target permutations have a probability density of zero, in expectation, which is then an upper bound on the number of inaccessible target permutations. This means the probability that all target permutations are accessible is then at least

$$1 - \tilde{n}!\sqrt{\frac{1}{2}\epsilon(N\tilde{n}/2, \tilde{n})}. \tag{82}$$

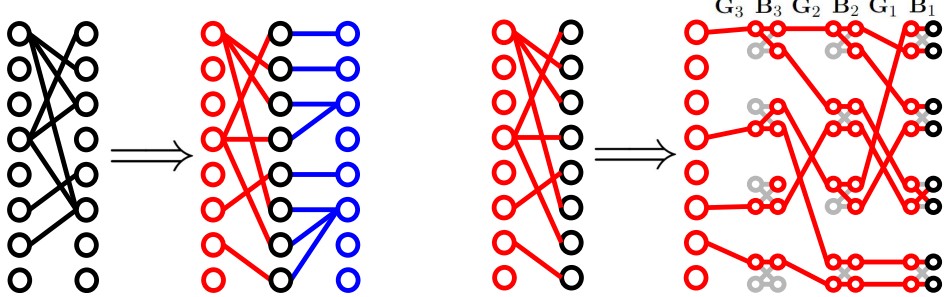

Figure 7: In this diagram, $\tilde{n} = 8$ and $p = 2$, for illustrative purposes. **Left:** Visual depiction of how a fanout forest followed by a fanin forest can represent arbitrary sparse connections. **Right:** Visual depiction of how $p + 1$ chained copies of $\tilde{G}(\boldsymbol{\theta})$ networks left-multiplied by block matrices diagonal can manifest a forest of fanout trees. $\tilde{G}_i$ are permutations implemented by $\tilde{G}(\boldsymbol{\theta})$ networks allowing for arbitrary connections, and $\boldsymbol{B}_i$ are block diagonal matrices which create the necessary fanouts. Data flows from left to right in the illustration, though successive operations are written out right to left when using mathematical notation. The largest fanout in this pattern of connections is 3, which is less than $2^p = 4$; all the fanins are 1.

Note that the leftmost $\boldsymbol{Q}_1$ in Equation 79 merely introduces a bijection between target permutations, and so does not change how many are accessible.

**Second part.** Suppose we would like to represent $p$ target permutations using $p$ independently generated $\tilde{G}(\boldsymbol{\theta})$ networks each of depth $M$. Equation 82 lower bounds the probability that each network can represent its respective permutation. Then the probability that all of the $p$ target permutations are accessible by their respective copies of $\tilde{G}(\boldsymbol{\theta})$ is union bounded by

$$1 - p\tilde{n}!\sqrt{\frac{1}{2}\epsilon(M\tilde{n}/2, \tilde{n})} \tag{83}$$

where the union is over the probability that each copy fails to represent its target permutation. Given that this succeeds, we now need to chain these permutations together with block diagonal matrices to represent arbitrary neural networks $\tilde{\boldsymbol{F}}$, with Equation 83 lower bounding the probability of failure.

Suppose we are successful in representing any combination of $p$ target permutations using $p$ distinct independently generated $\tilde{G}(\boldsymbol{\theta})$ networks of depth $M$. Then, since each $\tilde{G}(\boldsymbol{\theta})$ network's final (leftmost) operation is a block diagonal matrix, applying an additional block diagonal matrix afterward does not affect the operations that $\tilde{G}(\boldsymbol{\theta})$ can represent, since the set of block diagonal matrices we use is closed under matrix multiplication. Importantly, each block matrix can be used to copy a value from one index into two or to add two values together to leave one, creating fanin or fanout in the network. Then, interleaving $p + 1$ chained copies of $\tilde{G}(\boldsymbol{\theta})$ with block diagonal matrices left-multiplied for a total depth of $(p+1)M$ therefore creates an aggregate operation which can still be represented by a single $\tilde{G}(\boldsymbol{\theta})$ network of depth $(p+1)M$. This aggregate operation has up to $\tilde{n}$ arbitrary connections from input nodes to output nodes, with either all fanin at most 1 and all fanout at most $2^p$, or all fanout at most 1 and all fanin at most $2^p$. This is done by building a forest of fanin/fanout trees like illustrated on the right side of Figure 7.

If we compose such a $\tilde{G}(\boldsymbol{\theta})$ network of depth $(p+1)M$ with fanout up to $2^p$ together with a $\tilde{G}(\boldsymbol{\theta})$ network of depth $(p+1)M$ with fanin up to $2^p$, then we may represent any sparse matrix structure with at most $\tilde{n}$ nonzero weights and maximum fanin and fanout at most $2^p$, using a $\tilde{G}(\boldsymbol{\theta})$ network of depth $2(p+1)M$. This construction is illustrated on the left side of Figure 7. We may adjust the final (leftmost) block diagonal matrix on the fanout side to change the values of the $\tilde{n}$ arbitrarily positioned weights in the desired sparse matrix. Therefore, any sparse matrix with at most $\tilde{n}$ weights and max node indegree and outdegree at most $k$ can be represented by a $\tilde{G}(\boldsymbol{\theta})$ of depth $2(\lceil \log_2 k \rceil + 1)M$.

Then, any linear neural network of depth at most $d$, at most $\tilde{n}$ weights per layer, and maximum fanin and fanout of at most $k$ can be represented by a $\tilde{G}(\boldsymbol{\theta})$ network of depth $2Md(\lceil \log_2 k \rceil + 1)$, by composition

of sparse matrices. The probability that all of this is successful is merely the probability that all the $p = 2Md(\lceil \log_2 k \rceil + 1)$ permutations can be represented, which by Equation 83 is at least

$$1 - 2\tilde{n}!Md(\lceil \log_2 k \rceil + 1)\sqrt{\frac{1}{2}\epsilon(M\tilde{n}/2, \tilde{n})}. \tag{84}$$

Thus in summary, if we randomly generate a $\tilde{G}(\boldsymbol{\theta})$ network of depth $N = 2Md(\lceil \log_2 k \rceil + 1)$ for fixed constants $k$, $d$, and $M$, $\tilde{n}$ neurons per layer, and block size $f = 2$, then there is a probability of at least

$$1 - \tilde{n}!N\sqrt{\frac{1}{2}\epsilon\left(M\tilde{n}/2, \tilde{n}\right)} = 1 - \tilde{n}!N\sqrt{\frac{1}{2}\epsilon\left(\frac{\tilde{n}N}{4d(\lceil \log_2 k \rceil + 1)}, \tilde{n}\right)} \tag{85}$$

that $\tilde{G}(\boldsymbol{\theta})$ can represent every possible linear neural network of input and output dimension $\leq \tilde{n}$, depth $d$, at most $\tilde{n}$ nonzero weights per layer, and max fanin/fanout of at most $k$. $\qquad\square$

## D   Hessian Learning Local Minima (LODO can Generalize)

Laurent & von Brecht (2018) gives a theorem showing that for dense linear neural networks on convex losses where each layer is at least as wide as the input or output layer, all local minima in the neural network weights are also global minima. If we simplify LODO by approximating the inverse Hessian with a full dense matrix $\boldsymbol{G}(\boldsymbol{\theta}_t)$, then this theorem applies to the Hessian approximation error $||\boldsymbol{I} - \boldsymbol{G}(\boldsymbol{\theta}_t)\boldsymbol{H}||_F^2$, which the rescaled error $||\boldsymbol{B}\boldsymbol{D}^{-1}||_F^2$ used in Section 4.1 is a proxy for. Thus we may expect that any inverse Hessian approximation which LODO could converge to is of similar high quality to the best possible inverse Hessian approximation.

## E   Hyperparameters

In every experiment, we tuned the hyperparameters of each optimizer using a genetic algorithm of 10 generations and 32 individuals per generation (16 individuals per generation for the Resnet CIFAR10 task). Each hyperparamter was rescaled using $x \mapsto \ln x$ if the hyperparameter was a learning rate and $x \mapsto 1 - \ln(1 - x)$ if the hyperparameter was a decay parameter, so that the genetic algorithm would operate in a more well-behaved hyperparameter space. Starting from the default hyperparameters, each generation's mean hyperparameters were added to some Gaussian noise to create mutated hyperparameters for each individual, where the standard deviation of the noise was generation-dependent and followed a specific schedule. Each individual performed a generation-dependent number of steps of optimization, also according to a schedule. The next generation's mean hyperparameters were chosen to be the mean hyperparameters of the better performing half of the previous generation, as judged by the average training loss during the last 10% of training. We also halved all the learning rates (equiv. initial learning rates for LODO versions) after tuning for the image generation task because tests showed the loss to diverge if training time horizons were longer than 8k steps. Since LODO is a random optimizer, we used a different randomization seed for every individual. Table 5 lists the parameters of the tuning schedule for each task. The tuned hyperparameters can be found in Table 6.

## F   Task Setup Details

### F.1   Noisy Quadratic Bowl Task Details

This section fully explains the details of the setup of the noisy quadratic bowl task of Section 5.1. This 100 parameter task consists of a quadratic bowl for its loss landscape. Using a random uniform orthogonal matrix $\boldsymbol{U}$ and a diagonal matrix $\boldsymbol{D}$ consisting of a geometric sequence of 100 values starting at 0.001 and ending at 1, the Hessian of the quadratic bowl is set to $\boldsymbol{H} = \boldsymbol{U}\boldsymbol{D}\boldsymbol{U}^T$, and the center is set to the origin. However, whenever the loss and gradient are evaluated, the center of the quadratic bowl is perturbed by a random offset distributed as i.i.d. normal in each dimension, with mean zero and covariance $v\boldsymbol{I}$ for some $v > 0$ which

Table 5: Noise and step number schedules for tuning the optimizers' hyperparameters using the genetic algorithm presented in Appendix E

| Iteration | Noisy Quadratic Bowl | | Rosenbrock Function | | Image Generation | | Resnet-18 CIFAR10 | |
|---|---|---|---|---|---|---|---|---|
| | Noise stddev | steps | Noise stddev | steps | Noise stddev | steps | Noise stddev | steps |
| 0 | 3 | 1k | 3 | 200 | 3 | 1k | 3 | 50 |
| 1 | 3 | 1k | 3 | 200 | 3 | 1k | 3 | 50 |
| 2 | 3 | 1k | 3 | 200 | 3 | 1k | 3 | 50 |
| 3 | 3 | 1k | 2.5 | 200 | 3 | 1k | 3 | 50 |
| 4 | 2 | 1.5k | 2 | 200 | 2 | 1.5k | 2 | 50 |
| 5 | 1.7 | 1.5k | 1.5 | 200 | 1.7 | 1.5k | 1.7 | 50 |
| 6 | 1.4 | 2k | 1 | 200 | 1.4 | 2k | 1.4 | 100 |
| 7 | 1.2 | 3k | 0.75 | 200 | 1.2 | 3k | 1.2 | 200 |
| 8 | 0.9 | 5k | 0.5 | 200 | 0.9 | 5k | 0.9 | 300 |
| 9 | 0.6 | 8k | 0.3 | 200 | 0.6 | 8k | 0.6 | 500 |

defaults to 1. The initialization for this task is set to the origin. Due to the random wandering of the center, the expected loss rises linearly over time—unless the optimizer acts to prevent this, driving the error towards a steady state distribution. The expected loss after infinitely many steps can then be taken as a measure of the quality of an optimizer. The optimal solution for this task is to select the current minimum of the quadratic bowl at every timestep (which is what the Newton method would do). Due to the movement of the minimum between steps and loss evaluations, we should still expect this strategy to a achieve nonzero loss which can be analytically calculated to be $7.412v$. Table 1 shows the long term performance of LODO in comparison to other optimizers and the optimal solution of the Newton method. Table 7 and Figure 8 show that the performance of LODO is insensitive to the noise variance, and that LODO outperforms competitors for all noise levels.

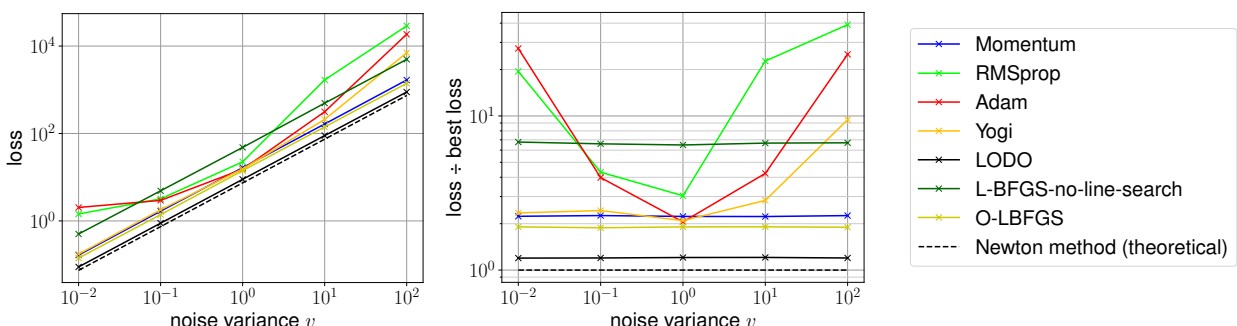

Figure 8: Average tracking error of the quadratic bowl minimum on the noisy quadratic bowl task of Section 5.1, with various different noise variances $v$. Values are averaged over the last 10% of training up to 300k steps. The theoretically best possible loss using Newton's method is also listed. The version of L-BFGS is one with stochastic modifications from (Schraudolph et al., 2007), instead of the original from (Nocedal & Wright, 1999). **Left:** Tracking error on a log-log plot. **Right:** Ratio of tracking error to the theoretically best possible tracking error by Newton's method, also on a log-log plot. Methods which choose steps that are equivariant to gradient scaling are insensitive to noise magnitude and show up as horizontal lines.

## F.2   CNN Image Generation Task Details

This section explains the CNN image generation task of Section 5.3, which is similar to the task of training a PixelCNN (Oord et al., 2016). Like PixelCNN, our CNN generates pixels row by row and column by column,

Table 6: Hyperparameters used for the experiments in Section 5, after tuning hyperparameters with a genetic algorithm as in Appendix E and halving the learning rates (equiv. initial learning rates for LODO variants) for the image generation task. $\beta$ and $\beta_1$ generally represent momentum decay rates while $\beta_2$ represent variance EMA decay rates. Dashes indicate that the optimizer was not used for that experiment.

| Optimizer | Hyperparameter | Value | | | |
|---|---|---|---|---|---|
| | | Noisy quadratic bowl | Rosenbrock function | Image generation | Resnet-18 CIFAR10 |
| Adam | Learning Rate | 1.164 | 0.9704 | 0.0009554 | 0.0004295 |
| | $\beta_1$ | 0.465 | 0.864 | 0.9323 | 0.9562 |
| | $\beta_2$ | 0.9884 | 0.99804 | 0.99505 | 0.99814 |
| Momentum | Learning Rate | 1.394 | 0.09870 | 0.003411 | 0.009942 |
| | $\beta$ | 0.529 | 0.9359 | 0.9640 | 0.99409 |
| RMSprop | Learning Rate | 0.449 | 0.004318 | $4.167 \times 10^{-5}$ | $9.877 \times 10^{-6}$ |
| | $\rho$ | 0.04943 | 0.02836 | 0.002258 | 0.03338 |
| | $\beta$ | 0.595 | 0.880 | 0.936 | 0.9394 |
| Yogi | Learning Rate | 2.169 | 0.2991 | 0.0009340 | 0.001829 |
| | $\beta_1$ | 0.4362 | 0.9273 | 0.9319 | 0.9730 |
| | $\beta_2$ | 0.9999723 | 0.998787 | 0.999708 | 0.9828 |
| LARS | Learning Rate | – | – | 0.0008552 | 0.002516 |
| | $\beta$ | – | – | 0.9343 | 0.9385 |
| | Weight Decay | – | – | $4.839 \times 10^{-5}$ | $5.370 \times 10^{-5}$ |
| L-BFGS | Learning Rate | 1.204 | – | – | – |
| | $\tau$ | 23002 | – | – | – |
| O-LBFGS | Learning Rate | 1.050 | – | – | – |
| | $\tau$ | 36721 | – | – | – |
| LODO | Meta-Learning Rate | 0.0009600 | 0.0001394 | $7.946 \times 10^{-6}$ | 0.0001134 |
| | $\beta$ | 0.195 | 0.897 | 0.9343 | 0.9490 |
| | Initial Learning Rate | 0.270 | 0.2946 | 0.08459 | 0.1725 |
| LODO-Diagonal | Meta-Learning Rate | – | – | 0.0005196 | – |
| | $\beta$ | – | – | 0.9860 | – |
| | Initial Learning Rate | – | – | 0.1325 | – |
| LODO-Global | Meta-Learning Rate | – | – | $7.951 \times 10^{-5}$ | – |
| | $\beta$ | – | – | 0.9525 | – |
| | Initial Learning Rate | – | – | 0.05583 | – |
| LODO-Residuals | Meta-Learning Rate | – | – | $3.829 \times 10^{-5}$ | – |
| | $\beta$ | – | – | 0.9301 | – |
| | Initial Learning Rate | – | – | 0.02134 | – |
| LODO-SGD | Meta-Learning Rate | – | – | 0.0007196 | – |
| | $\beta$ | – | – | 0.9499 | – |
| | Initial Learning Rate | – | – | 0.1035 | – |

and classifies the brightness of each pixel into 256 classes with crossentropy loss. Our data preprocessing is as follows. An MNIST image randomly selected, and one pixel location is chosen uniformly at random and blackened. All pixels below, or to the right of and in the same row as the selected pixel are blackened. The input into the CNN consists of this partially masked/blackened image (divided by 256 for normalization), an image indicating which pixels are specifically masked/blackened (indicated by $-1$ and 1), another image indicating which pixels are left of the selected pixel (indicated by $-1$ and 1), an image of a linear gradient from $-1$ to 1 in the x direction, and the same for the y direction. The last three images are present purely to break horizontal and vertical translation symmetries in the data, which has been shown to be helpful in

Table 7: Average tracking error of the quadratic bowl minimum on the noisy quadratic bowl task of Section 5.1, with various different noise variances $v$. Values are are averaged over the last 10% of training up to 100k steps. The theoretically best possible loss using Newton's method is also listed. The version of L-BFGS is one with stochastic modifications from (Schraudolph et al., 2007), instead of the original from (Nocedal & Wright, 1999). **Upper:** Loss values. **Lower:** LODO's average inverse Hessian approximation error $\sigma = \sqrt{||\boldsymbol{I} - \boldsymbol{G}(\boldsymbol{\theta}_t)\boldsymbol{H}||_F^2/n}$. $\sigma^2$ is measured by the unbiased estimator $\frac{1}{100}\sum_{i=1}^{100}||(\boldsymbol{I} - \boldsymbol{G}(\boldsymbol{\theta}_t)\boldsymbol{H})\boldsymbol{v}_i||_2^2$ with random independent unit vectors $\boldsymbol{v}_i$. Evidently, the ability of LODO to learn the inverse Hessian does not depend on the amount of noise.

| Noise variance $v$ | 0.01 | 0.1 | 1 | 10 | 100 |
|---|---|---|---|---|---|
| | loss at noise variance $v$ | | | | |
| Adam (Kingma & Ba, 2014) | 2.03 | 2.95 | 15.23 | 314.05 | 18602.71 |
| Momentum | 0.17 | 1.67 | 16.52 | 164.95 | 1672.73 |
| RMSprop (Hinton et al., 2012) | 1.44 | 3.20 | 22.50 | 1683.13 | 28973.22 |
| Yogi (Zaheer et al., 2018) | 0.17 | 1.80 | 15.51 | 209.83 | 7002.78 |
| L-BFGS (Schraudolph et al., 2007) | 0.50 | 4.89 | 48.04 | 494.04 | 4962.88 |
| O-LBFGS (Schraudolph et al., 2007) | 0.14 | 1.40 | 14.13 | 141.46 | 1405.51 |
| **LODO (ours)** | **0.09** | **0.89** | **8.94** | **89.54** | **887.94** |
| Newton Method (Optimal) | 0.07 | 0.74 | 7.41 | 74.12 | 741.18 |
| | $\sigma$ at noise variance $v$ | | | | |
| LODO | 0.7444 | 0.7421 | 0.7490 | 0.7479 | 0.7533 |

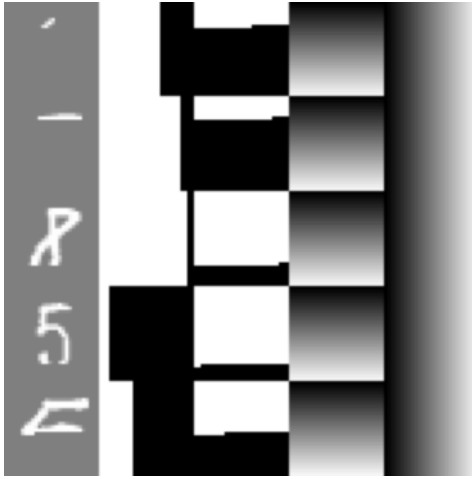

Figure 9: Batch of 5 tensors of preprocessed MNIST images from the image generation task of Section 5.3, each as one of the 5 rows of the image. Shown in each of 5 columns are the masked input, the left-of-selected-pixel indicator, visible mask, and two gradient images. The images in each row are concatenated together into a $28 \times 28 \times 5$ data tensor and then used as input into the CNN.

vision tasks involving collection of information from specific locations of an image specified by the data (Liu et al., 2018). The preprocessed data is visualized in Figure 9.

The CNN architecture we use for autoregression consists of:

- 5 input channels as previously described.

- Residual connection to Point A.

  - One pixel of zero padding, and convolution with 3 by 3 filters to 20 channels, with no activation.

- Point A.

- The following indented items are repeated 5 times.

  – The following indented items are repeated 4 times.

    ∗ Residual connection to Point B.

      · Convolution with 1 by 1 filters to 40 channels, with arctangent activation. We use the arctangent to mitigate gradient norm issues.

      · One pixel of zero padding, and three different depthwise convolutions concatenated together, with 3 by 3 filters to 120 channels, with arctangent activation.

      · Convolution with 1 by 1 filters to 20 channels, with no activation.

    ∗ Point B.

  – Average pooling of 2 by 2 blocks to halve the image size, with a pixel of zero padding beforehand if the image size is odd.

- Convolution with 1 by 1 filters to 256 channels, with softmax activation.

The loss is then the average crossentropy between true brightness of the query pixel and the distribution given by the softmax output of this CNN, over a batch size of 256 images. The whole CNN has about 94696 parameters, which are initialized with LeCun normal initialization (Klambauer et al., 2017). The task for the optimizer is to train these parameters to minimize this loss.

Figure 10 shows some imitation MNIST images generated using this CNN by sampling pixels one by one, after training with various optimizers including ours. The generated images are lower in quality because training was limited to 300k steps to best compare the efficiency of the optimizers.

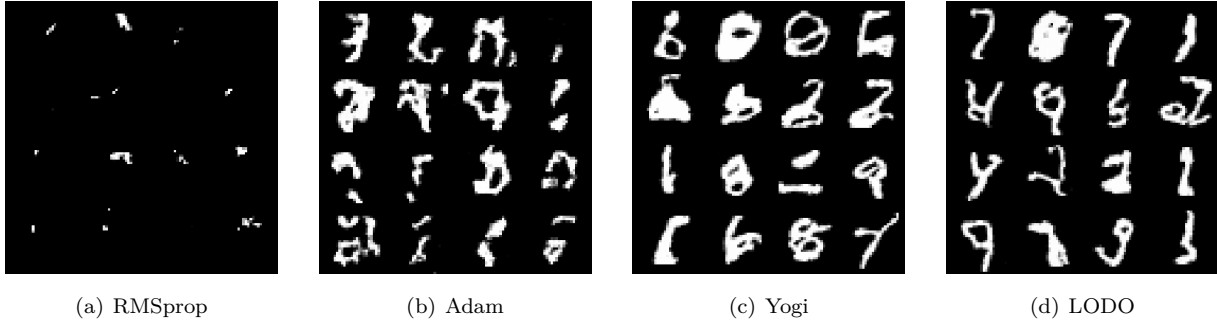

(a) RMSprop       (b) Adam       (c) Yogi       (d) LODO

Figure 10: Grids of 16 imitated MNIST images generated the image generation CNN of Section 5.3, trained for 300k steps using RMSprop, Adam, Yogi, and LODO.

Table 8: Mean losses between steps 180-200 while training on the Rosenbrock function minimization task, with various optimizers.

| Optimizer | Mean loss between steps 180-200 |
| --- | --- |
| Adam | 0.00005342 |
| RMSprop | 0.0008967 |
| Momentum | 0.01397 |
| Yogi | 0.0007916 |
| LODO (ours) | $0.001040 \pm 0.00002140$ |

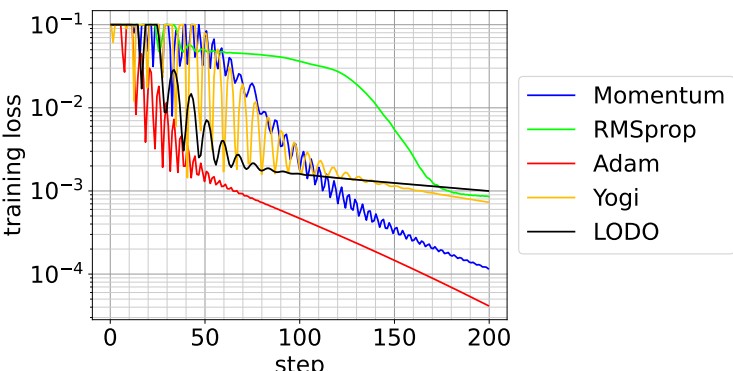

Figure 11: Log loss as a function of step, when using various optimizers on the Rosenbrock function minimization task.

## G  Rosenbrock Function Minimization Experiment

We probe the behavior of LODO with a small test task of finding the minimum of a rescaled Rosenbrock function $f(x, y) = 0.01(x-1)^2 + (x^2-y)^2$, which has no local minima and one global minimum at $(x, y) = (1, 1)$. We initialized the optimizers at $(x, y) = (-0.5, 2)$ and gave them 200 steps to run. The trajectory taken by LODO, shown in Figure 3, is similar to the short timescale dynamics of other optimizers using momentum, in that it tends to overshoot and then correct itself in an oscillatory manner. Learning curves in Figure 11 and losses in Table 8 show the performance of all the optimizers on this task.

## H  Training Loop Timing Details

This section describes how we timed each optimizer to report performance at specified times and step per second training speeds. We performed all optimization runs in TensorFlow 2, each with 40 Intel Xeon Gold 6248 CPUs and 2 Nvidia Volta V10 GPUs. Time reported includes all training time (forward and backward propagation, optimizer computation, data loading, preprocessing, etc.) except it does not include time taken to evaluate metrics such as the Hessian approximation error and the validation and test losses and accuracies.

