# OpenReview forum: "Learning to Optimize Quasi-Newton Methods"
_TMLR — Accepted by TMLR_

### Review · Reviewer_4FcD · 2023-05-15

**Summary Of Contributions:**

The paper proposed a novel pipeline, learning to optimize during optimization (LODO), to leverage a neural network to online meta-learn the best preconditioner during optimization. Both theoretical analysis and empirical studies are provided to support the effectiveness of LODO.

**Audience:**

Yes

**Broader Impact Concerns:**

This paper does not arouse any ethical concerns from my perspective.

**Claims And Evidence:**

Yes

**Requested Changes:**

- It would be good if the authors can provide some concrete examples to demonstrate why training on the fly is selected in the paper. For instance, comparing training on the flying and training an optimizer under some distribution.
- Comparison with other neural optimizers are necessary. If possible, conducting some large-scale experiments like image classification on ImageNet would make the paper stronger.

**Strengths And Weaknesses:**

Strengths:
- It is a novel direction to extend the idea of "learning to optimize" to more complicated optimization algorithms, in particular quasi-Newton method in this paper.
- A comprehensive and rigorous theoretical analysis of preconditioners parameterized as neural networks is presented in the paper, which has been rarely discussed in other relevant works.

Weaknesses:
- It was not clear to me what are the benefits of training a preconditioner on the fly. Normally, an optimizer which can generalize to different tasks is more desired. Training a specific optimizer for the target task would introduce much more computing.
- Evaluation was not very complete. Currently all selected tasks are relatively small-scaled and it is not clear whether LODO still works under large-scale training. On the other hand, only traditional hand-design optimizers are compared, which is not fair. At least other neural optimizers that learn the whole update rule should be included to show advantages of only learning preconditioners.

---

> ### Author Response · Authors · 2023-07-08
> **Response to Reviewer 4FcD**
>
> Dear Reviewer **4FcD**,
>
> Thank you for your valuable review, your constructive criticism has greatly contributed to the improvement of our paper. We have carefully addressed each of your concerns in our general response above. To quickly locate our responses specific to your comments, please refer to the following sections:
>
> 1. Why do we Train On The Fly?
> 2. Additional Comparisons Against Standard Quasi-Newton and Learned Optimizers
> 3. Rigorous Large-Scale Experimentation
>
> In these sections, you will find additional experiments, clarifications, and other changes that we have incorporated based on your feedback. We have uploaded an improved version of our paper, with updates in blue.
>
> We sincerely hope that our response adequately addresses your concerns and provides the necessary clarification. We kindly request you to consider our response when making your final decision regarding our paper.
>
> Thank you once again for your valuable input.
>
> Sincerely,
>
> The Authors

---

### Review · Reviewer_ZQcU · 2023-05-26

**Summary Of Contributions:**

This paper presents a new optimizer (LODO) based on the following principles:
- Parameterizing the preconditioner matrix inexpensively via EUNN parameterization
- Compute the gradient of the loss with respect to the parameters of the preconditioner, and update via taking a gradient step (or using e.g. Adam) on these parameters

The authors show that under some (semi-reasonable) assumptions, this estimation scheme converges to the true inverse Hessian preconditioner on quadratics.

Overall, I think this is a reasonable contribution to the world of quasi-Newton methods. There are a few issues which should be fixed before acceptance, however.

**Audience:**

Yes

**Claims And Evidence:**

Yes

**Requested Changes:**

- The most important change is to investigate LODO on more standard neural network training tasks. This will enable better comparison with other methods, and substantially broaden the appeal of the paper. I believe these experiments are critical for paper acceptance.
- There are several small experiments recommended above (eg comparison on estimation to quasi-Newton/Gauss-Newton estimation algorithms) that will improve the paper.


**Strengths And Weaknesses:**

## Strengths
- Overall, I think investigating gradient-based adaptation of Hessian approximations (or more task-specific targets than simple estimation error) is a worthwhile avenue of investigation.
- This paper shows the method can work, although the experiments do not show it outperforming simpler optimizers.
- The theory is reasonable for backing up the method.

## Weaknesses
- While the authors present their method as a learned optimization/L2O/meta-optimization method, I think this is misleading in comparison to other learned optimizers, which are much more expressive. This method is more reasonably placed in the class of quasi-Newton methods, although the authors may define QN methods as only those which directly aim to minimize Hessian/inverse Hessian estimation error.
- The authors should evaluate the performance of LODO on more standard neural network optimization problems. While I think the image generation model is interesting, evaluating the model on e.g. a standard resnet training would be much more valuable to the community. Ideally, I recommend the authors look at the MLCommons Algorithmic Efficiency benchmark: https://github.com/mlcommons/algorithmic-efficiency
- The Hessian estimation on varying problems should be evaluated, and better comparisons should be made to other quasi-Newton methods. This is especially true as the Hessian approximation error in Fig 2 seems to remain quite large, and it would be valuable to see how classic QN methods perform. Moreover, the Hessian approximation error should be compared as a function of the noise covariance, which will likely have different impact for different QN methods. The authors should also compare, in addition to classical QN methods like BFGS, to standard Gauss-Newton Hessian estimation methods (eg Levenberg-Marquardt).
- Section 5.3.1, in which alternative architectures are explored, is unclear and could be expanded. Generally, the biasing the permutation matrices toward the identity matrix seems like a reasonable approach that (slightly) improves performance; there are several extensions that could also be investigated, and potentially a more general class of preconditioner architectures can be presented.
- The assumption of b_{t+1} = A_{t_0} b_t - s_t feels quite limiting, and undermines the theoretical results. However, I don't have a good suggestions for improving the theoretical development, so I am fine to keep it as is. That being said, the validity of the theoretical results in based on A_t updating slowly, and showing that the dynamics under an updated A and a non-updated A are similar would validate the claims of section 4.

Some typos:
- Lemma B.1: "Let us A and B"
- Table 1 caption: "Vales are are averaged"

---

> ### Author Response · Authors · 2023-07-08
> **Response to Reviewer ZQcU**
>
> Dear Reviewer **ZQcU**,
>
> Thank you for your valuable review, your constructive criticism has greatly contributed to the improvement of our paper. We have carefully addressed each of your concerns in our general response above. To quickly locate our responses specific to your comments, please refer to the following sections:
>
> 1. L2O vs Quasi-Newton Interpretation
> 2. Rigorous Large-Scale Experimentation
> 3. Additional Comparisons Against Standard Quasi-Newton and Learned Optimizers
> 4. Noise Sensitivity
> 5. Alternative Architectures and Extensions
> 6. Static Preconditioner Approximation over Short Distances
>
> In these sections, you will find additional experiments, clarifications, and other changes that we have incorporated based on your feedback. We have uploaded an improved version of our paper, with updates in blue.
>
> We sincerely hope that our response adequately addresses your concerns and provides the necessary clarification. We kindly request you to consider our response when making your final decision regarding our paper.
>
> Thank you once again for your valuable input.
>
> Sincerely,
>
> The Authors

---

### Review · Reviewer_biMo · 2023-06-25

**Summary Of Contributions:**

This paper introduces a new optimizer that uses a meta-learning method to learn a pre-conditioner with a hypernetwork.  They theoretically show that the proposed optimizer approximates the inverse Hessian and empirically verifies it with synthetic data. The paper is well-motivated, and the empirical results show that the proposed method outperforms existing methods.

**Audience:**

Yes

**Broader Impact Concerns:**

I do not have any broader impact concerns.

**Claims And Evidence:**

Yes

**Requested Changes:**

See the weakness.

**Strengths And Weaknesses:**

Pros:
1) The paper is well-motivated and easy to follow.
2) It contains both theoretical analysis and empirical results, especially for the inverse Hessian problems.


Cons:
1) Except for the synthetic data, the paper only works on image generalization applications on mnist dataset.
2) Why is image generalization problem only? How about image classification? Can we have more experiments on more tasks?
3) Can we do larger-scale experiments on datasets and networks? Like Cifar10 or ImageNet with some modern networks, like resnet or transformer. If we can't, why it is hard? It is due to the computation cost? It seems not that high to me.
4) Does the performance boost come from the optimizer or does it work like some regularization? Can we have an experiment for fix the hyper networks (G(\theta)), and only learn the model parameters?

My main concern is that why we would like to use this method as an optimizer as it lacks use cases.

---

> ### Author Response · Authors · 2023-07-08
> **Response to Reviewer biMo**
>
> Dear Reviewer **biMo**,
>
> Thank you for your valuable review, your constructive criticism has greatly contributed to the improvement of our paper. We have carefully addressed each of your concerns in our general response above. To quickly locate our responses specific to your comments, please refer to the following sections:
>
> 1. Rigorous Large-Scale Experimentation
> 2. Why do we Train On The Fly?
>
> In these sections, you will find additional experiments, clarifications, and other changes that we have incorporated based on your feedback. We have uploaded an improved version of our paper, with updates in blue.
>
> We sincerely hope that our response adequately addresses your concerns and provides the necessary clarification. We kindly request you to consider our response when making your final decision regarding our paper.
>
> Thank you once again for your valuable input.
>
> Sincerely,
>
> The Authors

---

### Author Response · Authors · 2023-07-08
**General Response to Reviewers, Continued**

**Why do we Train On The Fly?**

Reviewer **4FcD** stated:

> It was not clear to me what are the benefits of training a preconditioner on the fly. Normally, an optimizer which can generalize to different tasks is more desired. Training a specific optimizer for the target task would introduce much more computing.

and requested the changes:

> It would be good if the authors can provide some concrete examples to demonstrate why training on the fly is selected in the paper. For instance, comparing training on the flying and training an optimizer under some distribution.

and Reviewer **biMo** also asked:

> Does the performance boost come from the optimizer or does it work like some regularization? Can we have an experiment for fix the hyper networks (G(\theta)), and only learn the model parameters?

In response to concerns about the importance of training on the fly and the behavior of the optimizer, we have trained LODO on the fly on one randomly sampled noisy quadratic bowl, and  then we froze $\mathbf{G}(\mathbf{\theta})$ and thus the resulting optimizer, using it to optimize on a different randomly sampled noisy quadratic bowl. When we do this, the frozen optimizer diverges to a loss of 4.998e+22 within the first ten steps when testing on a new quadratic bowl. If we learn the optimizer on the fly, we learn $\mathbf{G}(\mathbf{\theta})$ and it gets closer to $\mathbf{H}^{-1}$, so the exponential convergence has a speed which increases over time. We are pushing $\mathbf{G}(\mathbf{\theta})$ to try to have the same eigenvectors of $\mathbf{H}$, with reciprocal eigenvalues which become very large (again, to make convergence faster on the current problem). The large eigenvalues of $\mathbf{G}(\mathbf{\theta})$ mean that if we transfer it to a different quadratic bowl with Hessian $\mathbf{H}'$, $\mathbf{G}(\mathbf{\theta})\mathbf{H}'$ can have huge eigenvalues, which is why training with frozen pretrained LODO on a new problem diverges quickly. We added a few sentences to the end of Section 5.1 to outline this experimental result:

> To test the importance for LODO to learn $\mathbf{G}(\theta_t)$ on the fly, we froze $\mathbf{G}(\theta_t)$ after having used LODO for 100k steps as before, and then tried to use this ``pretrained'' LODO on a new quadratic bowl. Within 10 steps, the loss diverged to $4.997 \times 10^{22}$, indicating that LODO is best trained on the fly rather than beforehand on a task distribution.

The Hessian is a quantity which is completely task-specific and changes throughout the trajectory in the loss landscape. Swapping the initializations of weights attached to two neurons in the same layer causes the Hessian's rows and columns to shuffle around, and so the Hessian is also completely initialization-specific too. Since LODO learns an approximation of the inverse Hessian, it absolutely needs to be able to adjust itself dynamically to the current conditions of whatever task it is optimizing for, including the current value of the parameters; this is why we must meta-train on the fly. Without meta-training on the fly, the off-diagonal parts of the Hessian cannot be reasonably learned, undermining the benefits LODO gets from quasi-Newton methodology. We would hesitate to even call this algorithm "LODO" anymore.

About the computing cost, LODO only takes 32\% more wall time to take the same number of steps as the fastest optimizer we compared against, in our MNIST image generation experiment. Keep in mind that this 32\% _includes_ the time taken to fully meta-train the optimizer from scratch and use it on the test task for $t$ steps, whereas the other non-learned optimizers are just being used for $t$ steps, so training on the fly does not add much computing.

Meta learning the optimizer on the fly is a quite different kind of meta learning than the typical idea of creating an optimizer in advance by meta-training using a task distribution. This confers numerous advantages, such as the lack of an expensive double loop, second order gradients for pretraining and the curated task distributions. (The last paragraph of Section 6 provides much more detail into how LODO differs from the typical L2O setup.)

**Miscellaneous**

- Fixed typo in hyperparameters table, Table 6 in Appendix: 0.009600 -> 0.0009600.
- Fixed typos listed by Reviewer **ZQcU**
- Added hyperparameters for Resnet-18 CIFAR10 problem into the hyperparameters table

**Closing Remarks**

We hope that our response has cleared the concerns the Reviewers may have. If there are any remaining concerns, we encourage the Reviewers to discuss them as this will help us to improve our paper further.

Sincerely,

The Authors

---

### Author Response · Authors · 2023-07-08
**General Response to Reviewers, Continued**

**L2O vs Quasi-Newton Interpretation**

Reviewer **ZQcU** stated:

> While the authors present their method as a learned optimization/L2O/meta-optimization method, I think this is misleading in comparison to other learned optimizers, which are much more expressive. This method is more reasonably placed in the class of quasi-Newton methods, although the authors may define QN methods as only those which directly aim to minimize Hessian/inverse Hessian estimation error.

It is not incredibly clear where to draw the line since LODO is really a blend between quasi-Newton methods and L2O/hypergradient methods, so we do not find it too useful to define a distinction here. Regardless, we have changed the title to "Learning to Optimize Quasi-Newton Methods". We agree that L2O methods are more expressive with respect to behavior with isolated parameters. We emphasize a separate interpretation of expressiveness: LODO allows for nearly maximally complex linear interactions between multiple parameters, while other L2O methods do not consider inter-parameter interactions.

**Alternative Architectures and Extensions**

Reviewer **ZQcU** stated:

> Section 5.3.1, in which alternative architectures are explored, is unclear and could be expanded. Generally, the biasing the permutation matrices toward the identity matrix seems like a reasonable approach that (slightly) improves performance; there are several extensions that could also be investigated, and potentially a more general class of preconditioner architectures can be presented.

We believe that possible extensions and potential for further development is essential to a good paper; otherwise the paper's research is leading to nowhere and therefore has reduced significance to the advancement of the field. We had to choose to punctuate our continual research with a paper at some point; with Theorem 4.2 and the analysis of Section 4.1 along with experimental support, we felt this was the most appropriate place to conclude our paper. Yes, more is (almost) always better, we believe the potential extensions would not add experimental support for our theory at the core of our paper, and so is best left for future work. Note that Reviewer **ZQcU** has also stated:

> Overall, I think this is a reasonable contribution to the world of quasi-Newton methods.

**Static Preconditioner Approximation over Short Distances**

Reviewer **ZQcU** has stated that:

> The assumption of b_{t+1} = A_{t_0} b_t - s_t feels quite limiting, and undermines the theoretical results. However, I don't have a good suggestions for improving the theoretical development, so I am fine to keep it as is. That being said, the validity of the theoretical results in based on A_t updating slowly, and showing that the dynamics under an updated A and a non-updated A are similar would validate the claims of section 4.

For this analysis, it is not necessary to only set $t_0$ once and assume $\\mathbf{A}\_t \\approx \\mathbf{A}\_{t\_0}$ for the whole duration of training; instead $t_0$ is reassigned to the value of $t$ every time $\mathbf{A}$ takes one Euler step to travel a small distance, to keep the approximation accurate. We have clarified this in the paper by adding the paragraph:

> One might ask how the approximation made by Equation 7 by taking $\\mathbf{A}\_t \\approx \\mathbf{A}\_{t\_0}$ continues to hold while $\mathbf{A}$ flows to zero. When given a time $t_0$, this approximation is only necessary to reach the conclusion that over short distances, $\mathbf{A}$ has the long-term flow rate/direction of Equation 10. After every Euler step on Equation 10, we may reinstantiate $t_0$ to the new value of $t$, such that the accuracy of the approximation $\\mathbf{A}\_t \\approx \\mathbf{A}\_{t\_0}$ is fully restored for Equation 7 to be accurate, and Equation 10 can be reached once again so the next Euler step can be taken. It is in this way that the movement of $\mathbf{A}$ follows Equation 10.

near the end of Section 4.1 in the paper.

---

### Author Response · Authors · 2023-07-08
**General Response to Reviewers, Continued**

**Noise Sensitivity**

Reviewer **ZQcU** wanted to know how performance depends on the magnitude of noise:

> The Hessian estimation on varying problems should be evaluated, and better comparisons should be made to other quasi-Newton methods. This is especially true as the Hessian approximation error in Fig 2 seems to remain quite large, and it would be valuable to see how classic QN methods perform. Moreover, the Hessian approximation error should be compared as a function of the noise covariance, which will likely have different impact for different QN methods. The authors should also compare, in addition to classical QN methods like BFGS, to standard Gauss-Newton Hessian estimation methods (eg Levenberg-Marquardt).

In response to this, we duplicated the noisy quadratic bowl experiment for noise covariance matrices $vI$ for $v\in\{0.01, 0.1, 1.0, 10.0, 100.0\}$ for all optimizers while keeping the hyperparameters the same. We added Figure 8 and Table 7, which show that while some other optimizers' noise-rescaled training losses $\ell/v$ are heavily dependent on the noise covariance $v$, LODO's noise-rescaled training loss does not, and LODO still consistently outperforms all the other optimizers for all $v$. We add an extra row in Table 7 that shows that the normalized Hessian approximation error $\sigma_v=\sqrt{||\mathbf{I}-\mathbf{G}(\mathbf{\theta}_t)\mathbf{H}/v||_F^2/n}$ after 100k steps remains approximately the same regardless of $v$.

Data from Table 7 is copied below. Losses from Table 7:

|  **Noise variance $v$**  | **0.01**     | **0.1**  | **1**   | **10**   | **100**    |
| --------------- | --------------------- | -------- | ------- | -------- | ---------- |
| **Optimizer**   |                       |          |         |          |            |
| Adam            | $2.03$                | $2.95$   | $15.23$ | $314.05$ | $18602.71$ |
| Momentum        | $0.17$                | $1.67$   | $16.52$ | $164.95$ | $1672.73$  |
| RMSprop         | $1.44$                | $3.2$    | $22.5$  | $1683.13$| $28973.22$ |
| Yogi            | $0.17$                | $1.8$    | $15.51$ | $209.83$ | $7002.78$  |
| L-BFGS          | $0.5$                 | $4.89$   | $48.04$ | $494.04$ | $4962.88$  |
| O-LBFGS         | $0.14$                | $1.4$    | $14.13$ | $141.46$ | $1405.51$  |
| **LODO (ours)** | $\mathbf{0.09}$       | $\mathbf{0.89}$ | $\mathbf{8.94}$ | $\mathbf{89.54}$ | $\mathbf{887.94}$ |
| **Newton Method (Optimal)** | $0.07$    | $0.74$   | $7.41$  | $74.12$  | $741.18$   |

LODO's Hessian approximation errors from Table 7:

|  **Noise variance $v$**  | **0.01**     | **0.1**  | **1**   | **10**   | **100**    |
| --------------- | --------------------- | -------- | ------- | -------- | ---------- |
| LODO | $0.7444$    | $0.7421$   | $0.7490$  | $0.7479$  | $0.7533$   |

We believe LODO is unaffected by noise magnitude because the meta-optimizer, Adam, has long term behavior invariant to gradient magnitudes (ignoring hyperparameter $\epsilon$), meaning LODO's step size behavior is equivariant to gradient magnitudes. This pairs well with the scaling properties of the noisy quadratic bowl problem, which then allow LODO to achieve identical noise-rescaled training loss regardless of noise magnitude. We also added a paragraph outlining these results near the end of Section 5.1.

---

### Author Response · Authors · 2023-07-08
**General Response to Reviewers, Continued**

**Additional Comparisons Against Standard Quasi-Newton and Learned Optimizers**

Reviewer **ZQcU** requested additional comparisons to classical quasi-Newton methods:

> There are several small experiments recommended above (eg comparison on estimation to quasi-Newton/Gauss-Newton estimation algorithms) that will improve the paper.

and Reviewer **4FcD** requested additional comparisons to learned optimizers:

> Evaluation was not very complete. Currently all selected tasks are relatively small-scaled and it is not clear whether LODO still works under large-scale training. On the other hand, only traditional hand-design optimizers are compared, which is not fair. At least other neural optimizers that learn the whole update rule should be included to show advantages of only learning preconditioners.

and requested the change:

> Comparison with other neural optimizers are necessary. If possible, conducting some large-scale experiments like image classification on ImageNet would make the paper stronger.

In response to this, we have added some additional experiments:

1. We added BFGS and Levenberg-Marquardt optimizers to the noisy quadratic bowl experiment, in Table 1. While they outperform against LODO on this small task, these more well-established quasi-Newton optimization algorithms use at least $O(N^2)$ memory cost (while LODO and other optimizers we compare against use at most $o(N\log N)$), and therefore BFGS and Levenberg-Marquardt are too costly for us to add to any of our neural network experiments where $N>\,\sim10^5$ with the compute available to us.

2. We have also included the neural optimizer from Andrychowicz et al. (https://arxiv.org/abs/1606.04474, the optimizer is henceforth called "L2LBGDBGD") for comparison against L2O optimizers, in Table 1 and Figure 2. While Reviewer **4FcD** would like us to compare LODO against learned optimizers on large-scale training, unfortunately we are unable to meta-learn the L2LBGDBGD optimizer for larger problems within the timeframe of the review period with the resources available to us, and the pretrained optimizer is not available to us. Therefore we ran the comparison for the noisy quadratic bowl instead.

---

### Author Response · Authors · 2023-07-08
**General Response to Reviewers**

**We Thank the Reviewers**

We would like to thank the Reviewers for the time and effort they have put into their constructive feedback, which has helped us to improve our paper. Below, we address some questions and concerns about our paper, and explain how we have implemented the changes requested by the Reviewers.

**Rigorous Large-Scale Experimentation**

Reviewer **ZQcU** stated:

> The authors should evaluate the performance of LODO on more standard neural network optimization problems. While I think the image generation model is interesting, evaluating the model on e.g. a standard resnet training would be much more valuable to the community. Ideally, I recommend the authors look at the MLCommons Algorithmic Efficiency benchmark: https://github.com/mlcommons/algorithmic-efficiency

and requested the change:

> The most important change is to investigate LODO on more standard neural network training tasks. This will enable better comparison with other methods, and substantially broaden the appeal of the paper. I believe these experiments are critical for paper acceptance.

This is reiterated by Reviewer **biMo**, who said:

> Except for the synthetic data, the paper only works on image generalization applications on mnist dataset.

> Why is image generalization problem only? How about image classification? Can we have more experiments on more tasks?

> Can we do larger scale experiments on datasets and networks? Like Cifar10 or ImageNet with some modern networks, like resnet or transformer. If we can't, why is it hard? Is it due to the computation cost? It seems not that high to me.

> My main concern is that why we would like to use this method as an optimizer as it lacks use cases.

Reviewer **4FcD** also expressed concerns about the lack of experimentation in large-scale training.

In response to this concern, we have added an experiment where we train a Resnet18 on CIFAR10 classification in Section 5.4. The performance of all the optimizers is outlined in Figure 6 and Table 4 (in the modified text). The table is also copied below:

| Optimizer | Training loss (3000 steps) | Training loss (24k sec.) | Test accuracy (3000 steps) | Test accuracy (24k sec.) | Steps / sec. |
| --- | --- | --- | --- | --- | --- |
| Adam | $0.326 \pm 0.008$ | $0.446 \pm 0.016$ | $0.873 \pm 0.002$ | $0.863 \pm 0.003$ | $0.0793 \pm 0.0003$ |
| Momentum | $0.386 \pm 0.010$ | $0.563 \pm 0.031$ | $0.881 \pm 0.006$ | $0.864 \pm 0.008$ | $0.0794 \pm 0.0002$ |
| RMSprop | $2.380 \pm 0.375$ | $2.077 \pm 0.191$ | $0.434 \pm 0.043$ | $0.466 \pm 0.040$ | $0.0794 \pm 0.0001$ |
| Yogi | $0.398 \pm 0.006$ | $0.552 \pm 0.022$ | $0.885 \pm 0.003$ | $0.869 \pm 0.005$ | $0.0791 \pm 0.0004$ |
| LARS | $0.296 \pm 0.026$ | $0.403 \pm 0.034$ | $0.872 \pm 0.006$ | $0.864 \pm 0.007$ | $0.0788 \pm 0.0003$ |
| LODO (ours) | $0.624 \pm 0.027$ | $1.377 \pm 0.110$ | $0.845 \pm 0.009$ | $0.674 \pm 0.035$ | $0.0316 \pm 0.0003$ |

We observe that LODO performs reasonably against the best-performing optimizers (Adam, Yogi, Momentum, and LARS), and noticeably outperforms RMSProp. The "$\geq O(n^2)$" family of optimizers cannot be run on tasks of this scale. Since there are many crucial regularization techniques in computer vision, such as data augmentation and weight decay, a more in-depth study of how LODO interfaces with them is a fruitful direction for future work.

---

### Decision · Action_Editors · 2023-08-25

**Recommendation:** Accept as is

**Comment:**

This paper introduces a new optimizer (LODO) that learns a preconditioner during training. The reviewers generally agreed that the paper was easy to read, the topic was important, and that the theoretical results were sufficient for the type of problem studied. The empirical results were found to be less convincing, and the reviewers requested a number of additional experiments. The new additions to the manuscript thoroughly addressed the reviewers' major concerns.

There remain questions about whether LODO is likely to be of use in practice. However, the paper never makes that claim, and instead frames the contribution as a stepping stone towards developing meta-training-free learned optimizers. This and the other claims in the paper are clearly supported by the evidence presented and as such I recommend acceptance.

**Audience:**

Yes, the topics of this paper will interest a sizable fraction of the community.

**Claims And Evidence:**

Yes, the claims are scoped narrowly and the evidence supports them.